# Foundation model of neural activity predicts response to new stimulus types

Eric Y. Wang[1,2], Paul G. Fahey[1,2,3,4,5], Zhuokun Ding[1,2,3,4,5], Stelios Papadopoulos[1,2,3,4,5], Kayla Ponder[1,2], Marissa A. Weis[6], Andersen Chang[1,2], Taliah Muhammad[1,2], Saumil Patel[1,2,3,4,5], Zhiwei Ding[1,2], Dat Tran[1,2], Jiakun Fu[1,2], Casey M. Schneider-Mizell[7], MICrONS Consortium*, R. Clay Reid[7], Forrest Collman[7], Nuno Maçarico da Costa[7], Katrin Franke[1,2,3,4,5], Alexander S. Ecker[6,8], Jacob Reimer[1,2], Xaq Pitkow[1,2,9], Fabian H. Sinz[1,2,6,10] & Andreas S. Tolias[1,2,3,4,5,11 ✉]

The complexity of neural circuits makes it challenging to decipher the brain's algorithms of intelligence. Recent breakthroughs in deep learning have produced models that accurately simulate brain activity, enhancing our understanding of the brain's computational objectives and neural coding. However, it is difficult for such models to generalize beyond their training distribution, limiting their utility. The emergence of foundation models[1] trained on vast datasets has introduced a new artificial intelligence paradigm with remarkable generalization capabilities. Here we collected large amounts of neural activity from visual cortices of multiple mice and trained a foundation model to accurately predict neuronal responses to arbitrary natural videos. This model generalized to new mice with minimal training and successfully predicted responses across various new stimulus domains, such as coherent motion and noise patterns. Beyond neural response prediction, the model also accurately predicted anatomical cell types, dendritic features and neuronal connectivity within the MICrONS functional connectomics dataset[2]. Our work is a crucial step towards building foundation models of the brain. As neuroscience accumulates larger, multimodal datasets, foundation models will reveal statistical regularities, enable rapid adaptation to new tasks and accelerate research.

Deep artificial neural networks (ANNs) have revolutionized neuroscience by modelling neural activity based on sensory input, behaviour and internal states[3–9]. Task-driven models, for instance, have provided valuable insights into the visual cortex, as their hidden representations often align with biological neural activity when trained on tasks such as object classification or predictive coding[10,11]. With increasing access to large-scale neuroscience datasets, data-driven models are surpassing task-driven approaches[12], enabling in silico experiments that systematically analyse neuronal representations and computational principles. In vision research, such approaches help to characterize neuronal selectivity[13,14] and tuning functions under natural conditions[15], generating hypotheses for closed-loop experiments such as inception loops[16]. This in silico–in vivo strategy addresses key challenges in neuroscience, including high-dimensional inputs, nonlinear processing and experimental constraints.

A major challenge in neural network modelling, however, is generalization beyond the original training distribution[17]. Models trained on natural videos predict responses well within that domain but struggle with synthetic or parametric stimuli[18]. Given the historical importance of parametric stimuli in vision research[19–21], it is crucial to develop functional models that generalize across stimulus domains. Recent advancements in artificial intelligence, particularly foundation models trained on vast datasets[1], offer a solution. These models capture robust, transferable representations that generalize to novel tasks, as seen in language models trained on diverse text corpora[22,23].

Inspired by these breakthroughs, we developed a foundation model of the mouse visual cortex trained on extensive data to predict neural activity from dynamic visual stimuli and behaviour. We recorded responses to ecological videos from approximately 135,000 neurons across multiple visual cortex areas in 14 awake, behaving mice. With a subset of these data, we trained a deep neural network on recordings from eight mice, producing a 'foundation core' that captured shared latent representations and predicted neuronal responses across mice and cortical areas. Models using this foundation core could be rapidly adapted to new mice with minimal data, outperforming individualized models trained end-to-end. These models excelled in predicting neuronal responses to both in-domain natural videos and out-of-domain

[1]Center for Neuroscience and Artificial Intelligence, Baylor College of Medicine, Houston, TX, USA. [2]Department of Neuroscience, Baylor College of Medicine, Houston, TX, USA. [3]Department of Ophthalmology, Byers Eye Institute, Stanford University School of Medicine, Stanford, CA, USA. [4]Stanford Bio-X, Stanford University, Stanford, CA, USA. [5]Wu Tsai Neurosciences Institute, Stanford University, Stanford, CA, USA. [6]Institute of Computer Science and Campus Institute Data Science, University of Göttingen, Göttingen, Germany. [7]Allen Institute for Brain Science, Seattle, WA, USA. [8]Max Planck Institute for Dynamics and Self-Organization, Göttingen, Germany. [9]Department of Electrical and Computer Engineering, Rice University, Houston, TX, USA. [10]Institute for Bioinformatics and Medical Informatics, University of Tübingen, Tübingen, Germany. [11]Department of Electrical Engineering, Stanford University, Stanford, CA, USA. *A list of authors and their affiliations appears at the end of the paper. ✉e-mail: tolias@stanford.edu

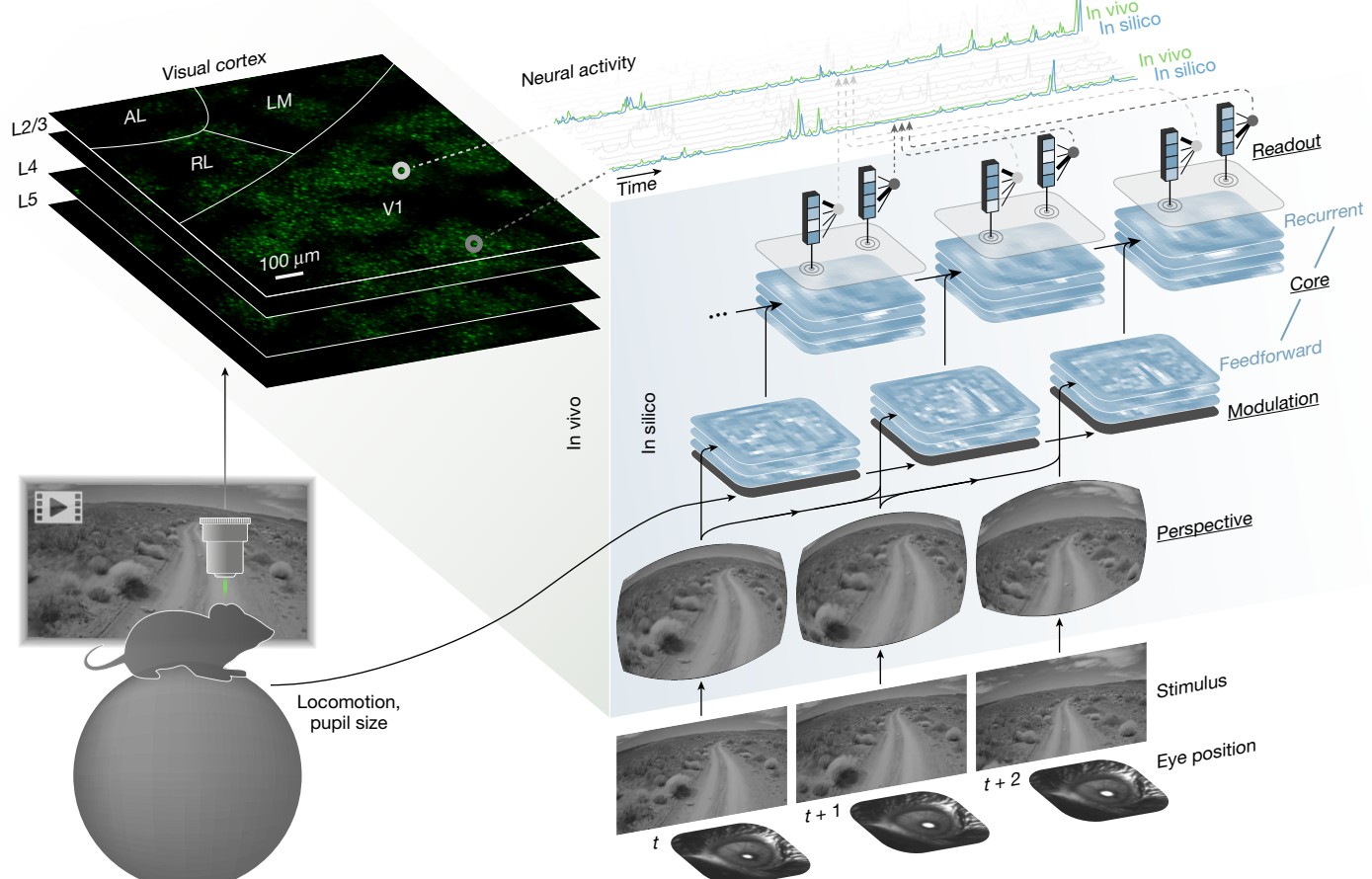

**Fig. 1 | ANN model of the visual cortex.** Top left, an in vivo recording session of excitatory neurons from several areas (V1, LM, RL and AL) and layers (layer 2/3 (L2/3), layer 4 (L4) and layer 5 (L5)) of the mouse visual cortex. Right, the architecture of the ANN model and the flow of information from inputs (visual stimulus, eye position, locomotion and pupil size) to outputs (neural activity). Underlined labels denote the four main modules of the ANN. For the modulation and core, the stacked planes represent feature maps. For the readout, the blue boxes represent the output features of the core at the readout position of the neuron, and the fanning black lines represent readout feature weights. The top of the schematic displays the neural activity for a sampled set of neurons. In vivo and in silico responses are shown for two example neurons. Stimulus adapted from Sports-1M Dataset (Andrej Karpathy; https://cs.stanford.edu/people/karpathy/deepvideo/); copyright 2014, IEEE, reprinted with permission from IEEE Proceedings, IEEE (CC BY3.0).

stimuli, including moving dots, flashing dots, Gabor patches, coherent noise and static natural images.

To evaluate the broader utility of our model, we assessed its ability to predict anatomical features. In the Machine Intelligence from Cortical Networks (MICrONS) dataset[2], which contains functional recordings and nanoscale anatomy of more than 70,000 neurons, our model accurately classified anatomically defined types of excitatory neurons. Furthermore, in other MICrONS studies, our model successfully predicted synaptic connectivity[24] and dendritic morphology[25].

In summary, we present a foundation model of neural activity that not only predicts visual cortex responses but also relates the functional properties of neurons to their anatomical features. Our results demonstrate the potential of data-driven foundation models to advance systems neuroscience by enabling scalable, generalizable representations of neural function.

## Dynamic functional model of the mouse visual cortex

To model the dynamic neuronal responses of the mouse visual cortex, we developed an ANN that comprised of four modules: perspective, modulation, core and readout (Fig. 1). The modular design enabled the ANN to accommodate diverse tasks and inputs. For instance, eye movements and different positioning of a mouse's head relative to the monitor can result in different perspectives of the same stimulus, despite best efforts

to limit experimental variability. To account for this, the perspective module of our ANN uses ray tracing and eye tracking data to infer the perspective of the mouse from the presented stimulus on the monitor (Extended Data Fig. 1). To account for behavioural factors that modulate the activity of the visual cortex[26], the modulation module transforms behavioural inputs (locomotion and pupil dilation) to produce dynamic representations of the mouse's behavioural and attentive state (Extended Data Fig. 2). The perspective and modulation modules provide visual and behavioural inputs, respectively, to the core module of the ANN. Composed of feedforward (3D convolution layers) and recurrent (long short-term memory) components, the core contains the majority of the modelling capacity of the ANN and produces nonlinear representations of vision that are modulated by behaviour. These representations are mapped onto the activity of individual neurons by the readout module, which performs a linear combination of the features generated by the core at one specific location, the neuron's receptive field. All four modules of the ANN (perspective, modulation, core and readout) were trained end-to-end to predict time series of neuronal responses to natural videos (details of model architecture and training are presented in Methods).

First, we evaluated the predictive accuracy of our ANN model architecture when trained on individual recording sessions lasting around 1 h. Predictive accuracy was measured by the correlation between the recorded and the predicted responses to a novel set of stimuli that were not included in model training. To account for in vivo noise,

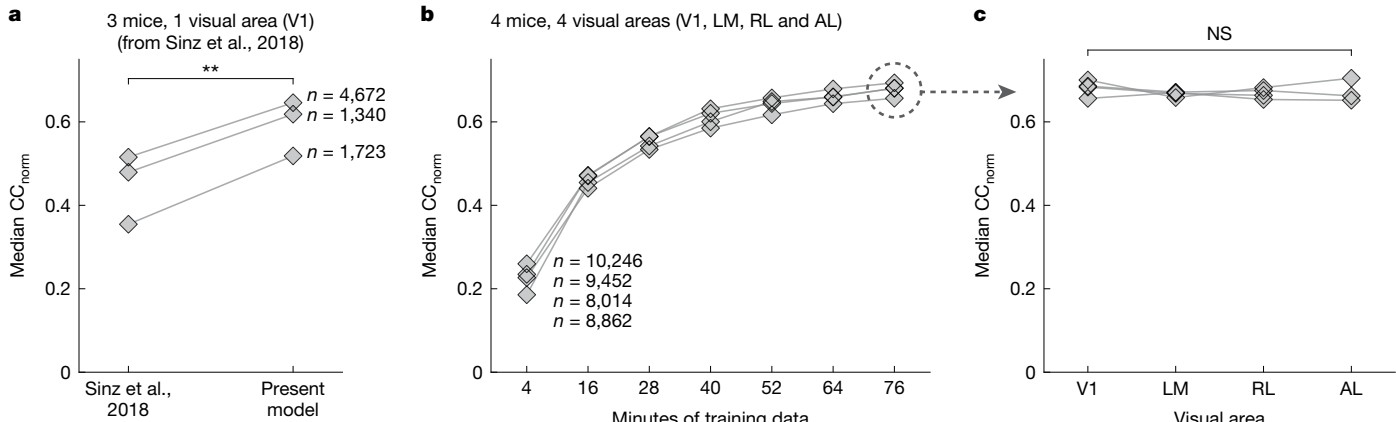

**Fig. 2 | Predictive accuracy of models trained on individual recording sessions. a**, Predictive accuracy (median $CC_{norm}$ across neurons; Methods) of our model versus a previous state-of-the-art dynamic model of the mouse visual cortex (Sinz et al.[18]). We trained and tested our model on the same set of data that were used in ref. 18—V1 neuronal responses to natural videos from three mice. $n$ refers to the number of neurons per mouse. **$P < 0.01$, paired two-way $t$-test, $t = 14.53$, d.f. = 2. **b**, Predictive accuracy of our models versus the amount of data used for training for four new recording sessions and mice.

For each recording session, training data were partitioned into 7 fractions ranging from 4 min to 76 min. Separate models (diamonds) were trained on the differing fractions of training data, and all were tested on the same held-out testing data. Models of the same mice are connected by lines. **c**, Predictive accuracy for each visual area from models that were trained on the full data. We did not find a statistically significant relationship between predictive accuracy and visual areas (linear mixed effects model[37]; NS, not significant by Wald test, $P = 0.45$, d.f. = 3).

the correlation was normalized by an estimated upper bound on the performance that could be achieved by a perfect model[27]. Using this normalized correlation coefficient ($CC_{norm}$) as the metric of predictive accuracy, we compared our model to the previous best-performing dynamic model of the mouse visual cortex[18]. Trained and tested on the same data from that study (dynamic primary visual cortex (V1) responses to natural videos), our model showed a 25–46% increase in predictive accuracy on held-out test data across the three recording sessions used in Sinz et al.[18] (Fig. 2a). This level of increase in performance is substantial for predictive models of the visual cortex. We also evaluated the predictive accuracy of our model on newly collected data that contained multiple visual areas (Fig. 2b). Of note, we found that the performance of our model for higher visual areas (lateromedial (LM), rostrolateral (RL) and anterolateral (AL)) was similar to V1 (Fig. 2c), despite the increased complexity of neuronal tuning to more complex features exhibited by higher visual areas[28,29].

Next, we performed lesion studies to determine the effect that individual components of the model had on predictive accuracy (Extended Data Fig. 3). Removing either of the 2 behavioural modules resulted in a modest but significant reduction in reduced predictive accuracy: 2.3% reduction for perspective (Extended Data Fig. 3a–e) and 2.8% for modulation (Extended Data Fig. 3f–j). For the core component, we found that using 3D convolutions in the feedforward component significantly improved performance compared to 2D convolutions, although the difference was small at 0.88% (Extended Data Fig. 3k–o). We also evaluated the objective function used for training and found that Poisson negative-log likelihood loss significantly outperformed mean squared error loss, with a performance difference of 9.6% (Extended Data Fig. 3p–t). In summary, our ANN model sets new standards for predicting dynamic neuronal responses of the visual cortex, with individual components contributing modest but significant improvements. Notably, the main driver of increased performance is the much larger dataset used for training (Fig. 2b), aligning with scaling laws and the observation that ANN performance in general improves with increasing data[30].

## Generalization to new subjects and stimulus domains

The remarkable performance of foundation models in other domains— for example, natural language[22] and image generation[23]—originates

from their vast quantities of training data. However, collecting large amounts of neuronal data from individual neurons and animals presents challenges. Individual recording sessions are limited in duration by experimental factors such as attentiveness and stability of the recording device. To overcome this limitation, we combined data from multiple recording sessions, resulting in a total of more than 900 min of natural video responses from 8 mice, 6 visual areas (V1, LM, AL, RL, anteromedial (AM) and posteromedial (PM)) and around 66,000 neurons (Extended Data Table 1). These data were used to train a single, shared ANN core (Fig. 3a) with the goal of capturing common representations of vision that underlie the dynamic neuronal response of the visual cortex for a representative set of neurons and a group of mice. This representation could then be used to fit models of new mice to improve their performance with limited data. Here we refer to the representative group of eight mice as the 'foundation cohort', the trained ANN component as the 'foundation core', and ANNs derived from the foundation core as 'foundation models'.

To evaluate the representation of the visual cortex captured by the foundation core, we froze its parameters and transferred it to ANNs with new perspective, modulation and readout components fitted to new mice (Fig. 3a). Each new mouse was shown an assortment of stimuli, designated for either model training or testing. The training stimuli consisted of natural videos, and we used different portions of this, spanning from 4 min to 76 min, to fit ANN components to the new mice. This approach aimed to examine the relationship between the models' performance and the amount of training data for each new mouse. The testing stimuli included natural videos that were not part of the training set (Fig. 3b'), new stimulus domains such as static natural images (Fig. 3c'), and four types of parametric stimuli (Fig. 3d'–g'), comprising drifting Gabor filters, flashing Gaussian dots, directional pink noise and random dot kinematograms. To test the role of the foundation core in prediction performance, we trained a set of control models that differed from the foundation models only by the core component. For these controls (individual models), all four components—core, perspective, modulation and readout—were trained end-to-end using training data from a single recording session. For the foundation models, training data from the new mice were used only to fit the perspective, modulation and readout components, and the core was trained on the foundation cohort as described above and was frozen (Fig. 3a).

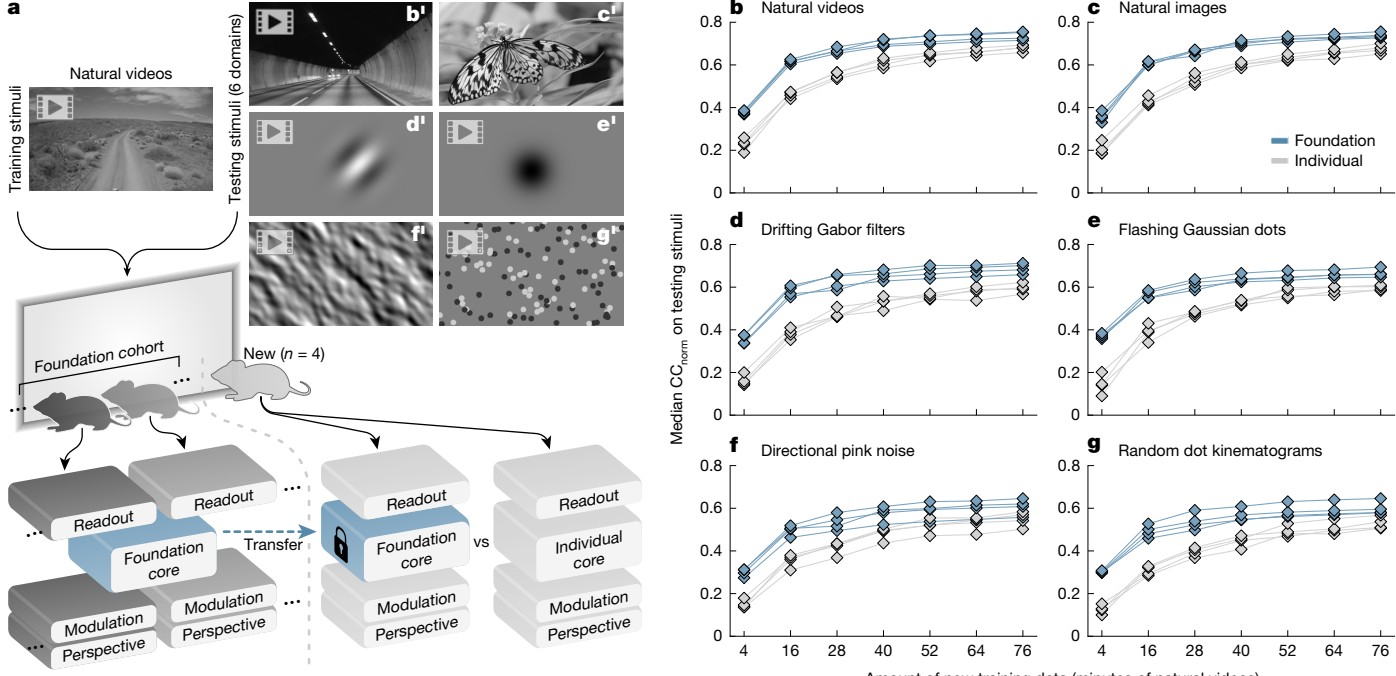

**Fig. 3 | Predictive accuracy of foundation models. a**, Schematic of the training and testing paradigm. Natural video data were used to train: (1) a combined model of the foundation cohort of mice with a single foundation core; and (2) foundation models versus individual models of new mice. Stimulus adapted from Sports-1M Dataset (Andrej Karpathy; https://cs.stanford.edu/people/karpathy/deepvideo/); copyright 2014, IEEE, reprinted with permission from IEEE Proceedings, IEEE (CC BY 3.0). **b′–g′**, The models of the new mice were tested with stimuli comprising natural videos (**b′**; adapted from Pixabay image (https://pixabay.com/photos/black-and-white-tunnel-the-way-1730543/; CC0 Content)), natural images (**c′**; adapted from Pixabay image (https://pixabay.com/photos/butterfly-insect-meadow-491166/; CC0 Content)), drifting Gabor filters (**d′**), flashing Gaussian dots (**e′**), directional pink noise (**f′**) and random dot kinematograms (**g′**). **b–g**, Corresponding plots for **b′–g′**, respectively, show the predictive accuracy (median $CC_{norm}$ across neurons) as a function of the amount of training data for foundation models versus individual models (grey) of the new mice (4 mice × 7 partitions of training data × 2 types of models = 56 models (diamonds)). Models of the same mouse and type (foundation or individual) are connected by lines. Number of neurons per mouse: 8,862, 8,014, 9,452 and 10,246, respectively.

When tested on natural videos, foundation models outperformed individual models and required less training data from the new mice to achieve high levels of predictive accuracy (Fig. 3b). For instance, individual models required more than 60 min of training data to surpass a median $CC_{norm}$ of 0.65 for all mice, whereas foundation models required less than 30 min (Fig. 3b). This performance gain was observed across all tested stimulus domains, including those that were in new stimulus domains (Fig. 3c′–g′)—that is, out-of-distribution from the training domain of natural videos (Fig. 3b′). Notably, no stimuli from the out-of-distribution domains were used to train any component of the models, including the foundation core. Nevertheless, foundation models were more accurate at predicting responses to new stimulus domains and required substantially less training data from the new mice (Fig. 3c–g). For example, when predicting drifting Gabor filters, the foundation models were able to achieve a performance of median $CC_{norm}$ greater than 0.55 using only 16 min of natural video training data. In contrast, the individual models required more than an hour of training data to reach the same performance level (Fig. 3d). This highlights the substantial difference in the data efficiency of these models—that is, the amount of training data (sample complexity) required from new subjects to accurately fit their neuronal responses. Thus, training a foundation dynamic core on natural video data pooled from multiple cortical layers, areas and mice produces a robust and transferable representation of the visual cortex that generalizes to new mice and improves model performance for natural videos and for novel stimulus domains.

When combining functional studies of the brain with other modalities such as anatomy, there is typically a limited amount of time available for in vivo recordings before destructive histological analysis is performed. Whereas traditionally this would limit the number of functional studies that can be performed in vivo, predictive models allow essentially unlimited scans to be performed in silico, even after tissue has been destroyed. To enable this for the MICrONS project, responses to natural videos were collected for the purpose of model training. Owing to the challenge of completing all 14 scans in the same mouse in as short a period as possible, the amount of training data collected from each experiment (mean 42 min, range 33–53 min, depending on optical quality and mouse behavioural profile) was less than in the other recording sessions described in this Article. With the available amount of data, individual models—with all components trained on a single experiment—achieved a median $CC_{norm}$ of 0.48–0.65 when tested on a held-out set of natural videos. By applying our foundation modelling paradigm—transferring the foundation core and fitting only the perspective, modulation and readout components on a single experiment—the median $CC_{norm}$ increased to 0.58–0.76 (Extended Data Fig. 4). This highlights the advantage of the foundation modelling approach when there is a limited amount of data available for training.

## Classical studies of parametric tuning

By leveraging the foundation core and transfer learning, we were able to create accurate foundation models for individual mice (Fig. 3). These models enable essentially unlimited in silico experiments for studying representations, testing theories and generating novel hypotheses that can be verified in vivo. Here we assessed the precision with which classical tuning properties of the visual cortex could be replicated at the individual neuronal level in our foundation model. We presented mice—not part of the foundation cohort—with natural video stimuli in order to train their ANN counterparts (Fig. 4a). Additionally, we

presented parametric stimuli (Fig. 4b',c') to measure the orientation, direction and spatial tuning of the recorded neurons. Subsequently, we presented the same parametric stimuli to the corresponding in silico neurons and measured their properties for comparison (Fig. 4b,c). This was done for 3 mice and approximately 30,000 neurons from 4 visual areas (V1, LM, AL and RL).

To measure orientation and direction tuning, we presented directional pink noise (Fig. 4b'), which encoded coherent motion of different directions (0–360°) and orientations (0–180°). First, we computed the strength of orientation and direction tuning via selectivity indices for orientation (OSI) and direction (DSI). There was a high correspondence between in vivo and in silico estimates for both OSI (Fig. 4d) and DSI (Fig. 4f), which validated the foundation model's estimates of tuning strength for orientation and direction. Next, we estimated the preferred angles of orientation and direction of neurons by fitting a directional parametric model (mixture of von Mises distributions) to the responses. For strongly tuned neurons, the in vivo and in silico estimates of preferred angles of orientation and direction were closely matched (Fig. 4e,g). For example, for strongly orientation-tuned neurons with an in silico OSI greater than 0.5 (11% of neurons), the median difference between the in vivo and in silico estimates of preferred orientation was 4°, and with a lower OSI threshold of over 0.3 (43% of neurons), the median difference was 7° (Fig. 4e).

To measure spatial tuning, we presented flashing Gaussian dots (Fig. 4c') to the neurons described above. We computed a spike-triggered average (STA) of the stimulus, which was used to estimate: (1) the strength of spatial tuning for Gaussian dots (non-uniformity of the STA) via the spatial selectivity index (SSI); and (2) the preferred location (peak of the STA) via least-squares fitting of the STA to a spatial parametric model (2D Gaussian distribution). Although using the Gaussian dot stimulus did not elicit strong SSI for the majority of neurons, for those neurons that were strongly tuned in silico, we observed a close match between in vivo and in silico estimates of spatial tuning strength, measured by SSI (Fig. 4h). For instance, for strongly tuned neurons with in silico SSI greater than 8, the median distance between the in vivo and in silico estimates of the preferred location was 0.02 of the monitor width (Fig. 4i), approximately 2° in visual space.

Together, these results demonstrate the accuracy of estimating tuning parameters for classical functional properties from our foundation model with no prior training on parametric stimuli. Therefore, rather than presenting parametric stimuli in vivo, parametric tuning can be performed in silico with an accurate and validated foundation model, freeing up valuable in vivo experimental time for other purposes.

## Prediction of structural properties of neurons

The function of the neocortex emerges mechanistically from its circuit structure. The MICrONS project, a landmark dataset in neuroscience, provides unprecedented scale and resolution, combining millimetre-scale functional recordings with anatomical structure at nanometre resolution, across multiple visual cortical areas of a single mouse. In the MICrONS mouse, the responses of more than 70,000 excitatory neurons to natural videos were measured across 14 sequential scans, encompassing a 1 mm³ volume spanning V1, LM, AL and RL visual areas. This volume was subsequently subjected to serial electron microscopy and dense morphological reconstruction (Fig. 5b), resulting in detailed structures of approximately 60,000 excitatory neurons and 500 million synapses, representing the largest integrated study of neocortical structure and function to date[2].

We used the foundation modelling paradigm to the MICrONS dataset to model the function of excitatory neurons within the 1 mm³ volume. The model's readout module maps the output of the foundation core onto individual neuronal responses. The readout parameters of each neuron consist of two components: readout position and readout feature weights (Fig. 5a). We trained readout parameters for all excitatory

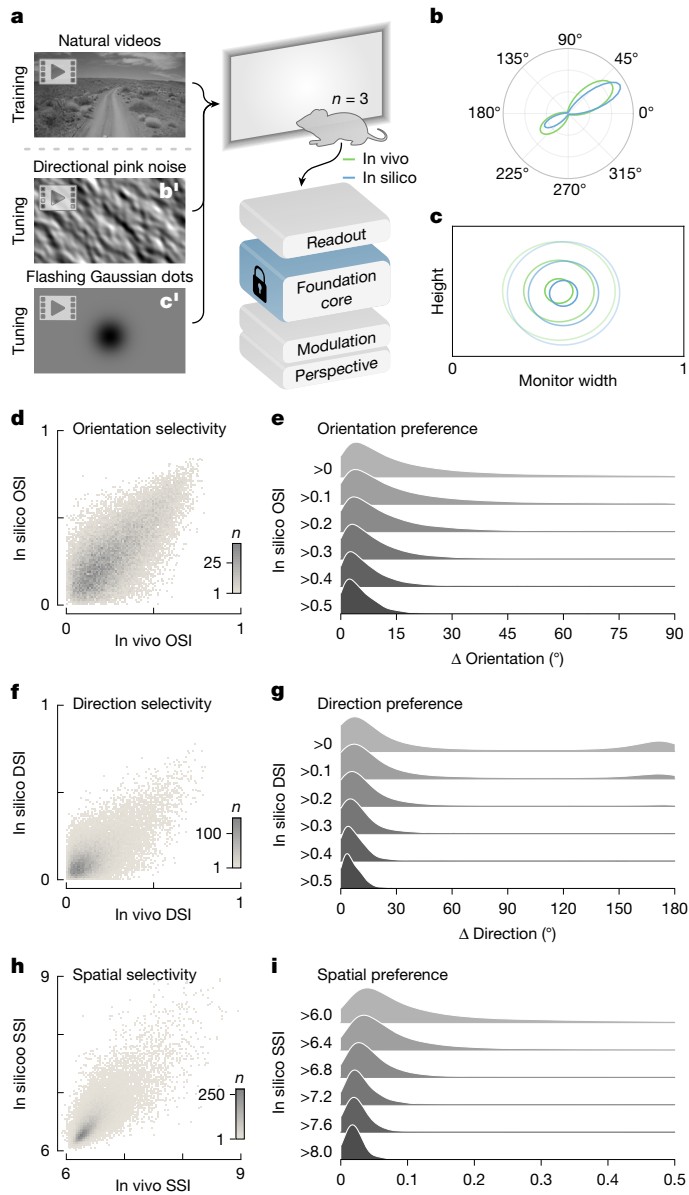

**Fig. 4 | Parametric tuning from foundation models. a,b',c',** Schematic of the experimental paradigm: foundation models of new mice ($n$ = 3) were trained with natural videos, and estimates of parametric tuning were computed from in vivo and in silico responses to synthetic stimuli (directional pink noise (**b'**) and flashing Gaussian dots (**c'**)). Adapted from Sports-1M Dataset (Andrej Karpathy; https://cs.stanford.edu/people/karpathy/deepvideo/); copyright 2014, IEEE, reprinted with permission from IEEE Proceedings, IEEE (CC BY 3.0). **b,c,** In vivo and in silico estimates of an example neuron's parametric tuning to orientation and direction (**b**) and spatial location (**c**). **d,f,h,** Binned scatter plots of in vivo and in silico estimates of orientation (**d**), direction (**f**) and spatial (**h**) selectivity indices. Grey colour bar indicates the number of neurons ($n$) in each bin. DSI, direction selectivity index; OSI, orientation selectivity index; SSI, spatial selectivity index. **e,g,i,** Density histograms of differences between in vivo and in silico estimates of preferred orientation (**e**), direction (**g**) and spatial location (**i**). Histograms containing increasingly selective groups of neurons thresholded by in silico OSI (**e**), DSI (**g**) and SSI (**i**) are stacked from top to bottom. Density histograms were produced via kernel density estimation using Scott's bandwidth.

neurons recorded in the MICrONS volume and we investigated whether these parameters would be useful for studying the structure–function relationship of the brain.

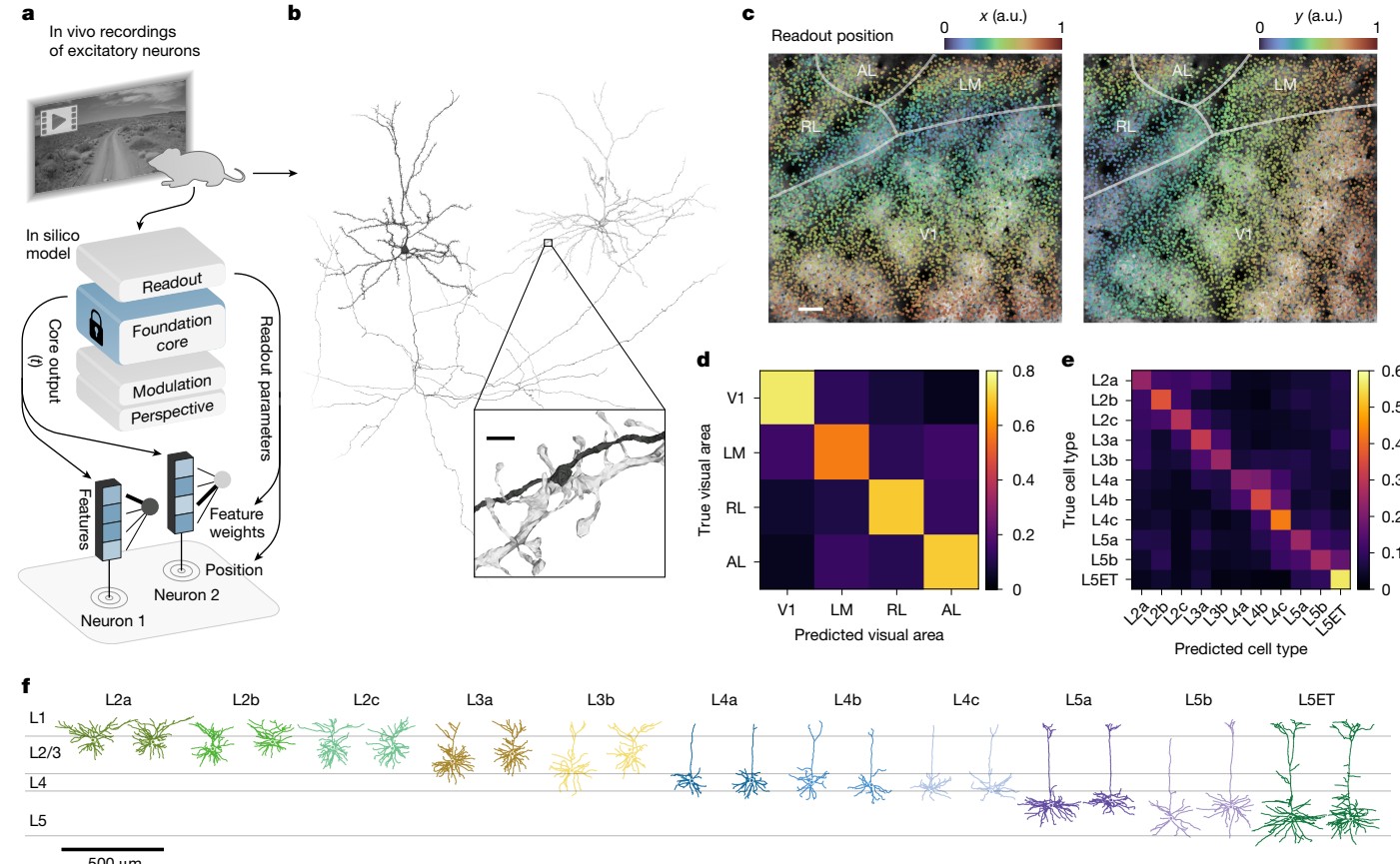

**Fig. 5 | The foundation model of the MICrONS volume relates neuronal function to structure and anatomy. a**, Schematic of a foundation model of the MICrONS mouse, trained on excitatory neuronal responses to natural videos. At the bottom, the readout at a single time point is depicted, showing the readout positions and feature weights for two example neurons. Adapted from Sports-1M Dataset (Andrej Karpathy; https://cs.stanford.edu/people/karpathy/deepvideo/); copyright 2014, IEEE, reprinted with permission from IEEE Proceedings, IEEE (CC BY 3.0). **b**, Meshes of two example neurons, reconstructed from serial electron microscopy. Inset, magnified view of the indicated area, showing a synapse between these two neurons, with the pre-synaptic axon in black and the post-synaptic dendrite in grey. Scale bar, 1 μm. **c**, Coloured scatter plots of readout positions of all neurons from a recording session of the MICrONS mouse, overlaid on a top-down view of the recording window with annotated visual areas (V1, LM, RL and AL) and boundaries. Plots are coloured by the $x$ (left) and $y$ (right) coordinates of the readout positions. Scale bar, 100 μm. a.u., arbitrary units. **d**, Confusion matrix of MICrONS visual areas predicted from readout feature weights, normalized per row. The diagonal represents the recall for each visual area. **e**, Confusion matrix of MICrONS excitatory neuron cell types predicted from readout feature weights, normalized per row. The excitatory neuron cell types are from Schneider-Mizell et al.[33]. The diagonal represents the recall for each cell type. **f**, Morphologies of different types of excitatory neurons. Two example neurons are shown for each excitatory neuron cell type. L5ET, layer 5 extratelencephalic-projecting neurons.

We first examined the readout position, which consists of two parameters per neuron: azimuthal ($x$) and altitudinal ($y$) locations, specifying the centre of the receptive field learned by the model for each neuron. Analysis of the readout positions revealed that they accurately captured the retinotopic organization of the visual cortex (Fig. 5c). In V1, readout $x$ positions aligned with the medial–lateral axis, and $y$ positions aligned with the rostral–caudal axis. At the border of V1 and LM/RL, there was an inversion of the axis for the $x$ readout position, demarcating the transition zone between these areas. This organization of readout positions according to anatomical locations aligns well with prior studies of retinotopic organization in the mouse visual cortex[31,32].

Next, we investigated how the readout weights, a 512-dimensional vector per neuron, could be used to predict anatomical properties such as the visual area and morphologically defined cell types. These readout weights serve as a functional barcode, encoding the tuning of the neuron to visual features produced by the core module at its readout position. We found that these functional barcodes captured differences between visual areas (V1, LM, AL and RL). Using logistic regression, the readout weights could predict visual areas with a balanced accuracy of 68%, exceeding the chance level of 25%

(Fig. 5d). We further explored the possibility of predicting 11 morphologically defined excitatory cell types from layers 2 to 5 of the neocortex (Fig. 5f), which were identified by Schneider-Mizell et al.[33]. Again, using logistic regression, we achieved a balanced accuracy of 32% for cell-type prediction, outperforming the chance baseline of 9% (Fig. 5e). Because these cell types are fairly well separated across cortical depth (Fig. 5f), it is possible that the classifier has learned to predict depth directly from the depth-varying signal-to-noise ratio of two-photon (2P) imaging. To control for this potential confound, we trained a classifier to predict cell types from 2P depth (reduced model) and compared to a second classifier provided with both 2P depth and readout feature weights (full model). We found that the full model significantly outperformed the reduced model in predicting cell types (likelihood ratio test, $P < 10^{-9}$), indicating that the readout feature weights contribute to classifier performance. Collectively, these results demonstrate that our foundation model captures both functional and structural properties of neurons, making it a valuable tool for analysing structure–function relationships within the MICrONS volume and studying mechanisms of computation within the visual cortex.

## Discussion

In this Article, we introduce a foundation model of the mouse visual cortex that achieves state-of-the-art performance at predicting dynamic neuronal responses across multiple visual areas, marking notable progress towards an accurate functional digital twin of the mouse visual system.

Beyond excelling in the natural video domain on which it was trained, our model accurately predicted responses to new stimulus domains such as noise patterns and static images. Its generalization performance on new stimulus domains highlights its ability to capture nonlinear transformations from image space to neuronal activity in the mouse visual cortex. The foundation core enabled accurate models of new mice to be fitted with limited training data, which outperformed models with cores that were individually trained for each mouse, underscoring the power of transfer learning to capture latent representations that explain neural activity across mice[34].

Notably, we also demonstrate the utility of our model for making predictions beyond neural activity—for example, in tasks related to anatomy and connectivity—which greatly enhances its utility as a foundation model of the brain[1]. Specifically, by transferring the foundation core, we built a digital twin of the MICrONS dataset, which enabled us to extract a functional barcode for each neuron—a vector embedding that describes the input–output function of visual response. Although the model was trained without anatomical information (that is, without electron microscopy data), the functional barcodes successfully predicted anatomical cell types identified in an accompanying Article that analyses cellular morphology from the MICrONS electron microscopy dataset[33].

The compact representation of neuronal function provided by the functional barcodes in our model was utilized in several other MICrONS studies examining the relationship between neuronal function and anatomy. In a study characterizing the morphological landscape of cortical excitatory neurons, the functional barcodes predict detailed features of dendritic morphology of layer 4 pyramidal neurons[25]. In another Article, the functional barcodes predict synaptic connectivity, beyond what could be explained by physical proximity of axons and dendrites[24].

In summary, the results presented here and in the accompanying Articles[24,25,35,36] that utilize our model demonstrate the power of the foundation modelling approach for neuroscience research. Its ability to uncover subtle patterns in neural organization, such as cellular morphology and synaptic connectivity, showcases the potential of the model for driving new insights in neuroscience. In large projects such as MICrONS, where dataset longevity is highly desirable, the strong generalization capabilities of our foundation model and its ability to perform tasks beyond the original training domain offer clear benefits. This extends the utility of the dataset beyond its initial scope, enabling researchers to explore questions that were not originally considered and facilitating discoveries in neural circuit organization.

Our work was inspired by recent breakthroughs in artificial intelligence, where foundation models[1] trained on massive data volumes have demonstrated remarkable generalization in many downstream tasks. Applied to neuroscience, the foundation modelling paradigm overcomes a major limitation of previous common approaches in which models are individually trained using data from a single experiment. The limited amount of data hinders the accuracy of models as they learn from scratch the complex nonlinearities of the brain, even though there is a great deal of similarity in how visual neurons respond. By contrast, foundation models combine data from multiple experiments, including data from many brain areas and subjects under high-entropy natural conditions, giving them access to a much larger and richer set of data; only the specific idiosyncrasies of each individual mouse and its neurons must be learned separately. In other words, the similarities between neurons and subjects can be leveraged to identify common features of the brain, producing a more unified and accurate model of the brain that is informed by multiple subjects rather than one.

Our present foundation model is just the beginning, as it only models parts of the mouse visual system under passive viewing conditions. By expanding this approach to encompass complex, natural behaviours in freely moving subjects, incorporating additional brain regions and cell types, the development of multimodal foundation neuroscience models offers a powerful new approach to deciphering the algorithms that underpin natural intelligence. As we accumulate more diverse multimodal data—encompassing sensory inputs, behaviours and neural activity across various scales, modalities and species, foundation neuroscience models will enable us to decipher the neural code of natural intelligence, providing unprecedented insights into the fundamental principles of the brain.

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

**MICrONS Consortium**

**Eric Y. Wang**[1,2]**, Paul G. Fahey**[1,2,3,4,5]**, Zhuokun Ding**[1,2,3,4,5]**, Stelios Papadopoulos**[1,2,3,4,5]**, Marissa A. Weis**[6]**, Andersen Chang**[1,2]**, Taliah Muhammad**[1,2]**, Saumil Patel**[1,2,3,4,5]**, Zhiwei Ding**[1,2]**, Dat Tran**[1,2]**, Jiakun Fu**[1,2]**, Casey M. Schneider-Mizell**[7]**, R. Clay Reid**[7]**, Forrest Collman**[7]**, Nuno Maçarico da Costa**[7]**, Alexander S. Ecker**[6,8]**, Jacob Reimer**[1,2]**, Xaq Pitkow**[1,2,9]**, Fabian H. Sinz**[1,2,6,10] **& Andreas S. Tolias**[1,2,3,4,5,11]

## Methods

### Neurophysiological experiments

MICrONS data in Fig. 5 were collected as described in the accompanying Article[2], and data in Fig. 2a were collected as described[18]. Data collection for all other figures is described below.

All procedures were approved by the Institutional Animal Care and Use Committee of Baylor College of Medicine. Fourteen mice (*Mus musculus*, 6 females, 8 males, age 2.2–4 months) expressing GCaMP6s in excitatory neurons via *Slc17a7*-Cre and Ai162 transgenic lines (recommended and shared by H. Zeng; JAX stock 023527 and 031562, respectively) were anaesthetized and a 4-mm craniotomy was made over the visual cortex of the right hemisphere as described previously[26,38]. Mice were allowed at least five days to recover before experimental scans.

Mice were head-mounted above a cylindrical treadmill and 2P calcium imaging was performed using Chameleon Ti-Sapphire laser (Coherent) tuned to 920 nm and a large field of view mesoscope[39] equipped with a custom objective (excitation NA 0.6, collection NA 1.0, 21 mm focal length). Laser power after the objective was increased exponentially as a function of depth from the surface according to: $P = P_0 \times e^{(z/L_z)}$, where $P$ is the laser power used at target depth $z$, $P_0$ is the power used at the surface (not exceeding 20 mW), and $L_z$ is the depth constant (220 μm). The highest laser output of 100 mW was used at approximately 420 μm from the surface.

The craniotomy window was leveled with regards to the objective with six degrees of freedom. Pixel-wise responses from a region of interest spanning the cortical window (>2,400 × 2,400 μm, 2–5 μm per pixel, between 100 and 220 μm from surface, >2.47 Hz) to drifting bar stimuli were used to generate a sign map for delineating visual areas[31]. Area boundaries on the sign map were manually annotated.

For 11 out of 15 scans (including four of the foundation cohort scans), our target imaging site was a 1,200 × 1,100 μm² area spanning L2–L5 at the conjunction of lateral V1 and 3 lateral higher visual areas: AL, LM and RL. This resulted in an imaging volume that was roughly 50% V1 and 50% higher visual area. This target was chosen in order to mimic the area membership and functional property distribution in the MICrONS mouse[2]. Each scan was performed at 6.3 Hz, collecting eight 620 × 1,100 μm² fields per frame at 2.5 μm per pixel *x*–*y* resolution to tile a 1,200–1,220 × 1,100 μm² field of view at 4 depths (2 planes per depth, 20–40 μm overlap between coplanar fields). The four imaging planes were distributed across layers with at least 45 μm spacing, with 2 planes in L2/3 (depths: 170–200 μm and 215–250 μm), 1 in L4 (300–325 μm) and 1 in L5 (390–420 μm).

For the remaining four foundation cohort scans, our target imaging site was a single plane in L2/3 (depths 210–220 μm), spanning all visual cortex visible in the cortical window (typically including V1, LM, AL, RL, PM and AM). Each scan was performed at 6.8–6.9 Hz, collecting four 630 μm width adjacent fields (spanning 2,430 μm region of interest, with 90 μm total overlap). Each field was a custom height (2,010–3,000 μm) in order to encapsulate visual cortex within that field. Imaging was performed at 3 μm per pixel.

Video of the eye and face of the mouse was captured throughout the experiment. A hot mirror (Thorlabs FM02) positioned between the left eye and the stimulus monitor was used to reflect an IR image onto a camera (Genie Nano C1920M, Teledyne Dalsa) without obscuring the visual stimulus. The position of the mirror and camera were manually calibrated per session and focused on the pupil. Field of view was manually cropped for each session. The field of view contained the left eye in its entirety, and was captured at ~20 Hz. Frame times were time stamped in the behavioural clock for alignment to the stimulus and scan frame times. Video was compressed using the Labview MJPEG codec with quality constant of 600 and stored the frames in AVI file.

Light diffusing from the laser during scanning through the pupil was used to capture pupil diameter and eye movements. A DeepLabCut model[40] was trained on 17 manually labelled samples from 11 mice to label each frame of the compressed eye video (intraframe only H.264 compression, CRF:17) with 8 eyelid points and 8 pupil points at cardinal and intercardinal positions. Pupil points with likelihood >0.9 (all 8 in 72–99% of frames per scan) were fit with the smallest enclosing circle, and the radius and centre of this circle was extracted. Frames with <3 pupil points with likelihood >0.9 (<1.2% frames per scan), or producing a circle fit with outlier >5.5× s.d. from the mean in any of the 3 parameters (centre *x*, centre *y*, radius, <0.2% frames per scan) were discarded (total <1.2% frames per scan). Gaps of ≤10 discarded frames were replaced by linear interpolation. Trials affected by remaining gaps were discarded (<18 trials per scan, <0.015%).

The mouse was head-restrained during imaging but could walk on a treadmill. Rostro-caudal treadmill movement was measured using a rotary optical encoder (Accu-Coder 15T-01SF-2000NV1ROC-F03-S1) with a resolution of 8,000 pulses per revolution, and was recorded at ~100 Hz in order to extract locomotion velocity. The treadmill recording was low-pass filtered with a Hamming window to remove high-frequency noise.

### Monitor positioning and calibration

Visual stimuli were presented with Psychtoolbox in MATLAB to the left eye with a 31.0 × 55.2 cm (height × width) monitor (ASUS PB258Q) with a resolution of 1,080 × 1,920 pixels positioned 15 cm away from the eye. When the monitor is centred on and perpendicular to the surface of the eye at the closest point, this corresponds to a visual angle of 3.8° cm⁻¹ at the nearest point and 0.7° cm⁻¹ at the most remote corner of the monitor. As the craniotomy coverslip placement during surgery and the resulting mouse positioning relative to the objective is optimized for imaging quality and stability, uncontrolled variance in skull position relative to the washer used for head-mounting was compensated with tailored monitor positioning on a six-dimensional monitor arm. The pitch of the monitor was kept in the vertical position for all mice, while the roll was visually matched to the roll of the head beneath the headbar by the experimenter. In order to optimize the translational monitor position for centred visual cortex stimulation with respect to the imaging field of view, we used a dot stimulus with a bright background (maximum pixel intensity) and a single dark square dot (minimum pixel intensity). Randomly ordered dot locations drawn from either a 5 × 8 grid tiling the screen (20 repeats) or a 10 × 10 grid tiling a central square (approximately 90° width and height, 10 repeats), with each dot presentation lasting 200 ms. For five scans (four foundation cohort scans, one scan from Fig. 4), this dot-mapping scan targeted the V1–RL–AL–LM conjunction, and the final monitor position for each mouse was chosen in order to maximize inclusion of the population receptive field peak response in cortical locations spanning the scan field of view. In the remaining scans, the procedure was the same, but the scan field of view spanned all of V1 and some adjacent higher visual areas, and thus the final monitor position for each mouse was chosen in order to maximize inclusion of the population receptive field peak response in cortical locations corresponding to the extremes of the retinotopic map. In both cases, the yaw of the monitor visually matched to be perpendicular to and 15 cm from the nearest surface of the eye at that position.

A photodiode (TAOS TSL253) was sealed to the top left corner of the monitor, and the voltage was recorded at 10 kHz and time stamped with a 10 MHz behaviour clock. Simultaneous measurement with a luminance meter (LS-100 Konica Minolta) perpendicular to and targeting the centre of the monitor was used to generate a lookup table for linear interpolation between photodiode voltage and monitor luminance in cd m⁻² for 16 equidistant values from 0–255, and 1 baseline value with the monitor unpowered.

At the beginning of each experimental session, we collected photodiode voltage for 52 full-screen pixel values from 0 to 255 for 1-s trials. The mean photodiode voltage for each trial was collected with an 800-ms boxcar window with 200-ms offset. The voltage was converted

to luminance using previously measured relationship between photodiode voltage and luminance and the resulting luminance versus voltage curve was fit with the function $L = B + A \times P^\gamma$ where L is the measured luminance for pixel value $P$, and the median $\gamma$ of the monitor was fit as 1.73 (range 1.58–1.74). All stimuli were shown without linearizing the monitor (that is, with monitor in normal gamma mode).

During the stimulus presentation, display frame sequence information was encoded in a three-level signal, derived from the photodiode, according to the binary encoding of the display frame (flip) number assigned in order. This signal underwent a sine convolution, allowing for local peak detection to recover the binary signal together with its behavioural time stamps. The encoded binary signal was reconstructed for >96% of the flips. Each flip was time stamped by a stimulus clock (MasterClock PCIe-OSC-HSO-2 card). A linear fit was applied to the flip time stamps in the behavioural and stimulus clocks, and the parameters of that fit were used to align stimulus display frames with scanner and camera frames. The mean photodiode voltage of the sequence encoding signal at pixel values 0 and 255 was used to estimate the luminance range of the monitor during the stimulus, with minimum values of approximately 0.005–1 cd m$^{-2}$ and maximum values of approximately 8.0–11.5 cd m$^{-2}$.

### Scan and behavioural data preprocessing
Scan images were processed with the CAIMAN pipeline[41], as described[2], to produce the spiking activity neurons at the scan rate of 6.3–6.9 Hz. The neuronal and behavioural (pupil and treadmill) activity were resampled via linear interpolation to 29.967 Hz, to match the presentation times of the stimulus video frames.

### Stimulus composition
We used dynamic libraries of natural videos[42] and directional pink noise (Monet) as described[2], and the static natural image library as described in Walker et al.[16].

Dynamic Gabor filters were generated as described[43]. We used a spatial envelope that had a s.d. of approximately 16.4° in the centre of the monitor. A 10-s trial consisted of 10 Gabor filters (each lasting 1 s) with randomly sampled spatial positions, directions of motion, phases, spatial and temporal frequencies.

Random dot kinematograms were generated as described[44]. The radius of the dots was approximately 2.6° in the centre of the monitor. Each 10-s trial contained 5 patterns of optical flow, each lasting 2 s. The patterns were randomly sampled in terms of type of optical flow (translation: up/down/right/left; radial: in/out; rotation: clockwise/anticlockwise) and coherence of random dots (50%, 100%).

The stimulus compositions of the MICrONS recording sessions is described in the accompanying Article[2]. For all other recording session, the stimulus compositions are listed in Extended Data Table 1.

### Neural network architecture
Our model of the visual cortex is an ANN composed of four modules: perspective, behaviour, core and readout. These modules are described in the following sections.

### Perspective module
The perspective module uses ray tracing to infer the perspective or retinal activation of a mouse at discrete time points from two input variables: stimulus (video frame) and eye position (estimated centre of pupil, extracted from the eye tracking camera). To perform ray tracing, we modelled the following physical entities: (1) topography and light ray trajectories of the retina; (2) rotation of the retina; (3) position of the monitor relative to the retina; and (4) intersection of the light rays of the retina and the monitor.

(1) We modelled the retina as a uniform 2D grid mapped onto a 3D sphere via an azimuthal equidistant projection (Extended Data Fig. 1a). Let $\theta$ and $\phi$ denote the polar coordinates (radial and angular, respectively) of the 2D grid. The following mapping produces a 3D light ray for point $(\theta, \phi)$ of the modelled retina:

$$\mathbf{l}(\theta, \phi) : \begin{bmatrix} \theta \\ \phi \end{bmatrix} \mapsto \begin{bmatrix} \sin\theta\cos\phi \\ \sin\theta\sin\phi \\ \cos\theta \end{bmatrix}.$$

(2) We used pupil tracking data to infer the rotation of the occular globe and the retina. At each time point $t$, a multilayer perceptron (MLP; with 3 layers and 8 hidden units per layer for the Conv-LSTM architecture and 3 layers and 16 hidden units per layer for the CvT-LSTM architecture) is used to map the pupil position onto the 3 ocular angles of rotation:

$$\text{MLP} : \begin{bmatrix} p_{xt} \\ p_{yt} \end{bmatrix} \mapsto \begin{bmatrix} \hat{\theta}_{xt} \\ \hat{\theta}_{yt} \\ \hat{\theta}_{zt} \end{bmatrix},$$

where the $p_{xt}$ and $p_{yt}$ are the $x$ and $y$ coordinates of the pupil centre in the frame of the tracking camera at time $t$, and $\hat{\theta}_{xt}$, $\hat{\theta}_{yt}$ and $\hat{\theta}_{zt}$ are the estimated angles of rotation of about the $x$ (adduction–abduction), $y$ (elevation–depression) and $z$ (intorsion–extorsion) axes of the occular globe at time $t$.

Let $\mathbf{R}_x$, $\mathbf{R}_y$, $\mathbf{R}_z \in \mathbb{R}^{3\times3}$ denote rotation matrices about $x$, $y$ and $z$ axes. Each light ray of the retina $\mathbf{l}(\theta, \phi)$ is rotated by the occular angles of rotation:

$$\hat{\mathbf{l}}(\theta, \phi, t) = \mathbf{R}_z(\hat{\theta}_{zt})\mathbf{R}_y(\hat{\theta}_{yt})\mathbf{R}_x(\hat{\theta}_{xt})\mathbf{l}(\theta, \phi),$$

producing $\hat{\mathbf{l}}(\theta, \phi, t) \in \mathbb{R}^3$, the ray of light for point $(\theta, \phi)$ of the retina at time $t$, which accounts for the gaze of the mouse and the rotation of the occular globe.

(3) We modelled the monitor as a plane with six degrees of freedom: three for translation and three for rotation. Translation of the monitor plane relative to the retina is parameterized by $\mathbf{m}_0 \in \mathbb{R}^3$. Rotation is parameterized by angles $\bar{\theta}_x, \bar{\theta}_y, \bar{\theta}_z$:

$$[\mathbf{m}_x \quad \mathbf{m}_y \quad \mathbf{m}_z] = \mathbf{R}_z(\bar{\theta}_z)\mathbf{R}_y(\bar{\theta}_y)\mathbf{R}_x(\bar{\theta}_x),$$

where $\mathbf{m}_x, \mathbf{m}_y, \mathbf{m}_z \in \mathbb{R}^3$ are the horizontal, vertical, and normal unit vectors of the monitor.

(4) We computed the line-plane intersection between the monitor plane and $\hat{\mathbf{l}}(\theta, \phi, t)$, the gaze-corrected trajectory of light for point $ij$ of the retina at time $t$:

$$\mathbf{m}(\theta, \phi, t) = \frac{\mathbf{m}_0 \cdot \mathbf{m}_z}{\hat{\mathbf{l}}(\theta, \phi, t) \cdot \mathbf{m}_z}\hat{\mathbf{l}}(\theta, \phi, t),$$

where $\mathbf{m}(\theta, \phi, t)$ is the point of intersection between the monitor plane and the light ray $\hat{\mathbf{l}}(\theta, \phi, t)$. This is projected onto the monitor's horizontal and vertical unit vectors:

$$m^x(\theta, \phi, t) = (\mathbf{m}(\theta, \phi, t) - \mathbf{m}_0) \cdot \mathbf{m}_x,$$
$$m^y(\theta, \phi, t) = (\mathbf{m}(\theta, \phi, t) - \mathbf{m}_0) \cdot \mathbf{m}_y,$$

yielding $m^x(\theta, \phi, t)$ and $m^y(\theta, \phi, t)$, the horizontal and vertical displacements from the centre of the monitor/stimulus (Extended Data Fig. 1b). To produce inferred activation of the retinal grid at $(\theta, \phi, t)$, we performed bilinear interpolation of the stimulus at the four pixels surrounding the line-plane intersection at $m^x(\theta, \phi, t)$, $m^y(\theta, \phi, t)$.

### Modulation module
The modulation module is a small long short-term memory (LSTM) network[45] that transforms behavioural variables—that is, locomotion

and pupil size—and previous states of the network, to produce dynamic representations of the behavioural state and arousal of the mouse.

$$\text{LSTM} : \begin{bmatrix} r_t \\ p_t \\ p_t' \end{bmatrix}, \mathbf{h}_{t-1}^m, \mathbf{c}_{t-1}^m \mapsto \mathbf{h}_t^m, \mathbf{c}_t^m,$$

where $r$ is the running or treadmill speed, $p$ is the pupil diameter, $p'$ is the instantaneous change in pupil diameter, and $\mathbf{h}^m, \mathbf{c}^m \in \mathbb{R}^6$ are the 'hidden' and 'cell' state vectors of the modulation LSTM network.

In the CvT-LSTM architectures, the modulation LSTM has the form:

$$\text{LSTM} : \begin{bmatrix} r_t \\ p_t \end{bmatrix}, \mathbf{h}_{t-1}^m, \mathbf{c}_{t-1}^m \mapsto \mathbf{h}_t^m, \mathbf{c}_t^m,$$

where $\mathbf{h}^m, \mathbf{c}^m \in \mathbb{R}^{16}$.

The hidden state vector $\mathbf{h}^m$ is tiled across space to produce modulation feature maps $\mathbf{H}_t^m$:

$$\mathbf{h}_t^m \in \mathbb{R}^C \rightarrow \mathbf{H}_t^m \in \mathbb{R}^{C \times H \times W},$$

where $C$, $H$ and $W$ denote channel, height and width, respectively, of the feature maps. These feature maps $\mathbf{H}_t^m$ serve as the modulatory inputs into the recurrent portion of the core module at time $t$.

## Core module

The core module—comprised of feedforward and recurrent components—transforms the inputs from the perspective and modulation modules to produce feature representations of vision modulated by behaviour.

First, the feedforward module transforms the visual input provided by the perspective module. For this we used DenseNet architecture[46] with three blocks. Each block contains two layers of 3D (spatiotemporal) convolutions followed by a nonlinearity (ELU for Conv-LSTM and GeLU for CvT-LSTM architectures[47]) and dense connections between layers. After each block, spatial pooling was performed to reduce the height and width dimensions of the feature maps. To enforce causality, we shifted the 3D convolutions along the temporal dimension, such that no inputs from future time points contributed to the output of the feedforward module.

Next, the recurrent module transforms the visual and behavioural information provided by the feedforward and modulation modules, respectively, through a group of recurrent cells. We used a convolutional LSTM (Conv-LSTM)[48] as the architecture for each recurrent cell. For each cell $c$, the formulation of the Conv-LSTM is shown below:

$$\mathbf{X}_t^c = W_1 * \mathbf{H}_t^f + W_1 * \mathbf{H}_t^m + \sum_{c'} W_1 * \mathbf{H}_{t-1}^{c'},$$
$$\mathbf{I}_t^c = \sigma(W_3 * \mathbf{X}_t^c + W_3 * \mathbf{H}_{t-1}^c + \mathbf{b}_i^c),$$
$$\mathbf{O}_t^c = \sigma(W_3 * \mathbf{X}_t^c + W_3 * \mathbf{H}_{t-1}^c + \mathbf{b}_o^c),$$
$$\mathbf{F}_t^c = \sigma(W_3 * \mathbf{X}_t^c + W_3 * \mathbf{H}_{t-1}^c + \mathbf{b}_f^c),$$
$$\mathbf{G}_t^c = \tanh(W_3 * \mathbf{X}_t^c + W_3 * \mathbf{H}_{t-1}^c + \mathbf{b}_g^c),$$
$$\mathbf{C}_t^c = \mathbf{F}_t^c \odot \mathbf{C}_{t-1}^c + \mathbf{I}_t^c \odot \mathbf{G}_t^c,$$
$$\mathbf{H}_t^c = \mathbf{O}_t^c \odot \tanh(\mathbf{C}_t^c),$$

where $\sigma$ denotes the sigmoid function, $\odot$ denotes the Hadarmard product, and $W_k *$ denotes a 2D spatial convolution with a $k \times k$ kernel. $\mathbf{H}_t^f$, $\mathbf{H}_t^m$ are the feedforward and modulation outputs, respectively, at time $t$, and $\mathbf{H}_{t-1}^{c'}$ is the hidden state of an external cell $c'$ at time $t - 1$. For cell $c$ at time $t$, $\mathbf{X}_t^c$, $\mathbf{C}_t^c$ and $\mathbf{H}_t^c$ are the input, cell and hidden states, respectively, and $\mathbf{I}_t^c$, $\mathbf{O}_t^c$, $\mathbf{F}_t^c$ and $\mathbf{G}_t^c$ are the input, output, forget and cell gates.

To produce the output of the core network, the hidden feature maps of the recurrent cells are concatenated along the channel dimension:

$$\mathbf{H}_t = \text{Concatenate}(\mathbf{H}_t^{c=1}, \mathbf{H}_t^{c=2}, \ldots).$$

In some Conv-LSTM model variants used in this work, the recurrent module additionally receives explicit spatial information about the visual stimulus. To do this, a spatial grid encoding the position of each feature map element within the visual field is concatenated to the feedforward features and modulatory vector before entering the Conv-LSTM.

Given the recent popularity and success of transformer networks[49], we explored whether adding the attention mechanism to our network would improve performance. We modified the Conv-LSTM architecture to incorporate the attention mechanism from the convolutional vision transformer (CvT)[50]. This recurrent transformer architecture, which we name CvT-LSTM, is described as follows:

$$\mathbf{X}_t^c = W_1 * \mathbf{H}_t^f + W_1 * \mathbf{H}_t^m + \sum_{c'} W_1 * \mathbf{H}_{t-1}^{c'},$$
$$\mathbf{Z}_t^c = W_3 * \mathbf{X}_t^c + W_3 * \mathbf{H}_{t-1}^c,$$
$$\mathbf{Q}_t^c = W_1 * \mathbf{Z}_t^c,$$
$$\mathbf{K}_t^c = W_1 * \mathbf{Z}_t^c,$$
$$\mathbf{V}_t^c = W_1 * \mathbf{Z}_t^c,$$
$$\mathbf{A}_t^c = \text{Attention}(\mathbf{Q}_t^c, \mathbf{K}_t^c, \mathbf{V}_t^c),$$
$$\mathbf{I}_t^c = \sigma(W_1 * \mathbf{A}_t^c + W_1 * \mathbf{Z}_t^c + \mathbf{b}_i^c),$$
$$\mathbf{O}_t^c = \sigma(W_1 * \mathbf{A}_t^c + W_1 * \mathbf{Z}_t^c + \mathbf{b}_o^c),$$
$$\mathbf{F}_t^c = \sigma(W_1 * \mathbf{A}_t^c + W_1 * \mathbf{Z}_t^c + \mathbf{b}_f^c),$$
$$\mathbf{G}_t^c = \tanh(W_1 * \mathbf{A}_t^c + W_1 * \mathbf{Z}_t^c + \mathbf{b}_g^c),$$
$$\mathbf{C}_t^c = \mathbf{F}_t^c \odot \mathbf{C}_{t-1}^c + \mathbf{I}_t^c \odot \mathbf{G}_t^c,$$
$$\mathbf{H}_t^c = \mathbf{O}_t^c \odot \tanh(\mathbf{C}_t^c),$$

where attention is performed over query $\mathbf{Q}_t^c$, key $\mathbf{K}_t^c$ and value $\mathbf{V}_t^c$ spatial tokens, which are produced by convolutions of the feature map $\mathbf{Z}_t^c$. The technique of using convolutions with the attention mechanism was introduced with CvT[50], and here we extend it by incorporating it into a recurrent LSTM architecture (CvT-LSTM).

We compare the performance of Conv-LSTM versus CvT-LSTM recurrent architecture in Extended Data Fig. 5. When trained on the full amount of data, Conv-LSTM performs very similarly to CvT-LSTM. However, Conv-LSTM outperforms CvT-LSTM when trained on restricted data (for example, 4 min of natural videos). This was consistent for all stimulus domains that were used to test model accuracy—natural videos (Extended Data Fig. 5a), natural images (Extended Data Fig. 5b), drifting Gabor filters (Extended Data Fig. 5c), flashing Gaussian dots (Extended Data Fig. 5d), directional pink noise (Extended Data Fig. 5e) and random dot kinematograms (Extended Data Fig. 5f). The performance difference under data constraints may be due a better inductive bias of the Conv-LSTM. Alternatively, it could be due to a lack of optimization of the CvT-LSTM hyperparameters, and a more extensive hyperparameter search may yield better performance.

## Readout module

The readout module maps the core's outputs onto the activity of individual neurons. For each neuron, the readout parameters are factorized into two components: spatial position and feature weights. For a neuron $n$, let $\mathbf{p}^n \in \mathbb{R}^2$ denote the spatial position $(x, y)$, and let $\mathbf{w}^n \in \mathbb{R}^C$ denote the feature weights for that neuron, with $C = 512$ being the number of channels in the core module's output. To produce the response of that neuron $n$ at time $t$, the following readout operation is performed:

$$\mathbf{h}_t^n = \text{Interpolate}(\mathbf{H}_t, \mathbf{p}^n),$$
$$r_t^n = \exp(\mathbf{h}_t^n \cdot \mathbf{w}^n + b^n),$$

where $\mathbf{h}_t^n \in \mathbb{R}^C$ is a feature vector that is produced via bilinear interpolation of the core network's output $\mathbf{H}_t \in \mathbb{R}^{C \times H \times W}$ (channels, height,

width), interpolated at the spatial position $\mathbf{p}^n$. The feature vector $\mathbf{h}_t^n$ is then combined with the feature weights $\mathbf{w}^n$ and a scalar bias $b^n$ to produce the response $r_t^n$ of neuron $n$ at time $t$.

Due to the bilinear interpolation at a single position, each neuron only reads out from the core's output feature maps within a $2 \times 2$ spatial window. While this adheres to the functional property of spatial selectivity exhibited by neurons in the visual cortex, the narrow window limits exploration of the full spatial extent of features during model training. To facilitate the spatial exploration of the core's feature maps during training, for each neuron $n$, we sampled the readout position from a 2D Gaussian distribution: $\mathbf{p}^n \sim \mathcal{N}(\mathbf{\mu}^n, \mathbf{\Sigma}^n)$. The parameters of the distribution $\mathbf{\mu}^n$, $\mathbf{\Sigma}^n$ (mean, covariance) were learned via the reparameterization trick[51]. We observed empirically that the covariance $\mathbf{\Sigma}^n$ naturally decreased to small values by the end of training, meaning that the readout converged on a specific spatial position. After training, and for all testing purposes, we used the mean of the learned distribution $\mathbf{\mu}^n$ as the single readout position $\mathbf{p}^n$ for neuron $n$.

In Extended Data Fig. 6, we examine the stability of the learned readout feature weights across different recording sessions. Due to the overlap between imaging planes, some neurons were recorded multiple times within the MICrONS volume. We found that the readout feature weights of the same neuron were more similar than feature weights of different neurons that were close in proximity, indicating that the readout feature weights of our model offer an identifying barcode of neuronal function that is stable across experiments.

### Four-head ensemble
In the implemented models, the modulation, core, and readout modules are independently parameterized across four heads (with shared perspective transform and readout grid), and predictions are obtained by averaging standardized log-responses across heads.

### Model training
The perspective, behaviour, core, and readout modules were assembled together to form a model that was trained to match the recorded dynamic neuronal responses from the training dataset. Let $y_t^i$ be the recorded in vivo response, and let $r_t^i$ be the predicted in silico response of neuron $i$ at time $t$. The ANN was trained to minimize the Poisson negative-log likelihood loss, $\sum_{it} r_t^i - y_t^i \log(r_t^i)$, via stochastic gradient descent with Nesterov momentum[52]. The ANN was trained for 200 epochs with a learning rate schedule that consisted of a linear warm up in the first 10 epochs, cosine decay[53] for 90 epochs, followed by a warm restart and cosine decay for the remaining 100 epochs. Each epoch consisted of 512 training iterations/gradient descent steps. We used a batch size of 5, and each sample of the batch consisted of 70 frames (2.33 s) of stimulus, neuronal and behavioural data.

### Model hyperparameters
We used a grid search to identify architecture and training hyperparameters. Model performances for different hyperparameters were evaluated using a preliminary set of mice. After optimal hyperparameters were identified, we used the same hyperparameters to train models on a separate set of mice, from which the figures and results were produced. There was no overlap in the mice and experiments used for hyperparameter search and the mice and experiments used for the final models, results, and figures. This was done to prevent overfitting and to ensure that model performance did not depend on hyperparameters that were fit specifically for certain mice.

### Model testing
We generated model predictions of responses to stimuli that were included in the experimental recordings but excluded from model training. To evaluate the accuracy of model predictions, for each neuron we computed the correlation between the mean in silico and in vivo responses, averaged over stimulus repeats. The average in vivo response aims to estimate the true expected response of the neuron. However, when the in vivo response is highly variable and there are a limited number of repeats, this estimate becomes noisy. To account for this, we normalized the correlation by an upper bound proposed by Schoppe et al.[27]. Using $\overline{\phantom{x}}$ to denote average over trials or stimulus repeats, the normalized correlation $\mathrm{CC_{norm}}$ is defined as follows:

$$\mathrm{CC_{norm}} = \frac{\mathrm{CC_{abs}}}{\mathrm{CC_{max}}},$$

$$\mathrm{CC_{abs}} = \frac{\mathrm{Cov}(\overline{r}, \overline{y})}{\sqrt{\mathrm{Var}(\overline{r})\mathrm{Var}(\overline{y})}},$$

$$\mathrm{CC_{max}} = \sqrt{\frac{N\mathrm{Var}(\overline{y}) - \overline{\mathrm{Var}(y)}}{(N-1)\mathrm{Var}(\overline{y})}},$$

where $r$ is the in silico response, $y$ is the in vivo response, and $N$ is the number of trials. $\mathrm{CC_{abs}}$ is the Pearson correlation coefficient between the average in silico and in vivo responses. $\mathrm{CC_{max}}$ is the upper bound of achievable performance given the in vivo variability of the neuron and the number of trials.

### Parametric tuning
To estimate parametric tuning, we presented parametric stimuli to the mice and the models. Specifically, we used directional pink noise parameterized by direction/orientation and flashing Gaussian blobs parameterized by spatial location. Orientation, direction and spatial tuning were computed from the recorded responses from the mice and the predicted responses from the models. This resulted in analogous in vivo and in silico estimates of parametric tuning for each neuron. The methods for measuring the tuning to orientation, direction, and spatial location are explained in the following sections.

### Orientation and direction tuning
We presented 16 angles of directional pink noise, uniformly distributed between $[0, 2\pi)$. Let $\overline{r}_\theta$ be the mean response of a neuron to the angle $\theta$, averaged over repeated presentations of the angle. The OSI and DSI were computed as

$$\mathrm{OSI} = \frac{|\sum_\theta \overline{r}_\theta \, e^{i2\theta}|}{\sum_\theta \overline{r}_\theta},$$

$$\mathrm{DSI} = \frac{|\sum_\theta \overline{r}_\theta \, e^{i\theta}|}{\sum_\theta \overline{r}_\theta},$$

that is, the normalized magnitude of the first and second Fourier components.

To determine the parameters for orientation and direction tuning, we used the following parametric model:

$$f(\theta|\mu, \kappa, \alpha, \beta, \gamma) = \alpha e^{\kappa\cos(\theta-\mu)} + \beta e^{\kappa\cos(\theta-\mu+\pi)} + \gamma,$$

which is a mixture of two von Mises functions with amplitudes $\alpha$ and $\beta$, preferred directions $\mu$ and $\mu + \pi$, and dispersion $\kappa$, plus a baseline offset of $\gamma$. The preferred orientation is the angle that is orthogonal to $\mu$ between $[0, \pi]$, that is, $(\mu + \pi/2) \bmod \pi$. To estimate the parameters $\mu, \kappa, \alpha, \beta, \gamma$ that best fit the neuronal response, we performed least-squares optimization, minimizing $\sum_\theta (f(\theta|\mu, \kappa, \alpha, \beta, \gamma) - r_\theta)^2$.

Parameters were estimated via least square optimization for both the in vivo and in silico responses. Let $\hat{\mu}$ and $\overline{\mu}$ be the angles of preferred directions estimated from in vivo and in silico responses, respectively. The angular distances between the in vivo and in silico estimates of preferred direction (Fig. 4g) and orientation (Fig. 4e) were computed as follows:

$$\Delta\mathrm{Direction} = \arccos(\cos(\hat{\mu} - \overline{\mu})),$$
$$\Delta\mathrm{Orientation} = \arccos(\cos(2\hat{\mu} - 2\overline{\mu}))/2.$$

## Spatial tuning

To measure spatial tuning, we presented 'on' and 'off' (white and black), flashing (300 ms) Gaussian dots. The dots were isotropically shaped, with a s.d. of approximately 8 visual degrees in the centre of the monitor. The position of each dot was randomly sampled from a 17 × 29 grid tiling the height and width of the monitor. We observed a stronger neuronal response for 'off' compared to 'on', and therefore we used only the 'off' Gaussian dots to perform spatial tuning from the in vivo and in silico responses.

To measure spatial tuning, we first computed the STA of the stimulus. Let $\mathbf{x} \in \mathbb{R}^2$ denote the spatial location (height and width) in pixels. The value of the STA at location $\mathbf{x}$ was computed as follows:

$$\bar{s}_{\mathbf{x}} = \frac{\sum_t |s_{\mathbf{x}t} - s_0| r_t}{\sum_t r_t},$$

where $r_t$ is the response of the neuron, $s_{\mathbf{x}t}$ is the value of the stimulus at location $\mathbf{x}$ and time $t$, and $s_0$ is the blank or grey value of the monitor.

To measure the spatial selectivity of a neuron, we computed the covariance matrix or dispersion of the STA. Again using $\mathbf{x} \in \mathbb{R}^2$ denote the spatial location (height and width) in pixels:

$$z = \sum_{\mathbf{x}} \bar{s}_{\mathbf{x}},$$

$$\bar{\mathbf{x}} = \sum_{\mathbf{x}} \bar{s}_{\mathbf{x}} \mathbf{x}/z,$$

$$\Sigma_{\mathrm{STA}} = \sum_{\mathbf{x}} \bar{s}_{\mathbf{x}} (\mathbf{x} - \bar{\mathbf{x}})(\mathbf{x} - \bar{\mathbf{x}})^{\mathrm{T}}/z.$$

The SSI, or strength of spatial tuning, was defined as the negative-log determinant of the covariance matrix:

$$\mathrm{SSI} = -\log|\Sigma_{\mathrm{STA}}|.$$

To determine the parameters of spatial tuning, we used least squares to fit the STA to the following parametric model:

$$f(\mathbf{x}|\boldsymbol{\mu}, \Sigma, \alpha, \gamma) = \alpha \exp\left(-\frac{1}{2}(\mathbf{x} - \boldsymbol{\mu})^{\mathrm{T}} \Sigma^{-1} (\mathbf{x} - \boldsymbol{\mu})\right) + \gamma,$$

which is a 2D Gaussian component with amplitude $\alpha$, mean $\boldsymbol{\mu}$, and covariance $\Sigma$, plus a baseline offset of $\gamma$.

From the in vivo and in silico responses, we estimated two sets of spatial tuning parameters. Let $\hat{\boldsymbol{\mu}}$ and $\bar{\boldsymbol{\mu}}$ be the means (preferred spatial locations) estimated from in vivo and in silico responses, respectively. To measure the difference between the preferred locations (Fig. 4i), we computed the Euclidean distance:

$$\Delta \mathrm{Location} = \|\hat{\boldsymbol{\mu}} - \bar{\boldsymbol{\mu}}\|.$$

## Anatomical predictions from functional weights

To predict brain areas from readout feature weights, we used all functional units in the MICrONS data from 13 scans that had readout feature weights in the model. We trained a classifier to predict brain areas from feature weights using logistic regression with nested cross validation. For each of the 10 folds, 90% of the data was used to train the model with another round of 10 fold cross validation to select the best L2 regularization weight. The best-performing model was used to test on the held-out 10% of data. Finally, all of the predictions were concatenated and used to test the performance of the classifier (balanced accuracy) and generate the confusion matrix. The confusion matrix was normalized such that all rows sum to 1, thus the diagonal values represent the recall of each class.

To predict cell types, the same functional data source was used as in the brain area predictions. Cell types were obtained from CAVEclient initialized with 'minnie65_public' and table 'aibs_metamodel_mtypes_v661_v2'. To associate a neuron's functional data with its cell type, we merged the cell types to a match table made by combining the manual and fiducial-based automatic coregistration described in MICrONS Consortium et al.[2]. Finally, because each neuron could be scanned more than once, and thus could have more than one functional read-out weight, we subset the data such that each neuron only had one readout weight according to its highest cc_max. Following this procedure, $n = 16,561$ unique electron microscopy neurons remained. Out of the 20 cell classes, all excitatory neuron classes in L2–5 were chosen (except L5NP, which had comparably fewer coregistered cells), leaving 11 classes: L2a, L2b, L2c, L3a, L3b, L4a, L4b, L4c, L5a, L5b and L5ET. To train the classifier using readout weights to predict cell types, logistic regression was used with the same nested cross validation procedure and performance metric as described in the brain area predictions.

For testing whether readout weights contributed to cell-type predictions beyond imaging depth, the 2P depth of each functional unit was obtained from a 2P structural stack (stack session 9, stack idx 19) wherein all imaging planes were registered[2]. This provided a common reference frame for all functional units. The two logistic regression models (depth versus depth + readout weights) were trained with all of the data, and the predicted probabilities and coefficients from the models were used to run the likelihood ratio test, where a P value less than 0.05 was chosen as the threshold for statistical significance.

## Reporting summary

Further information on research design is available in the Nature Portfolio Reporting Summary linked to this article.

## Data availability

All MICrONS data are available at https://bossdb.org/project/microns-minnie. Further details are available at https://www.microns-explorer.org/cortical-mm3.

## Code availability

The source code and foundation model weights are available at https://github.com/cajal/fnn. The model training and analysis pipeline can be accessed at https://github.com/cajal/foundation. The experimental recording and calcium imaging pipeline can be accessed at https://github.com/cajal/pipeline.

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

**Acknowledgements** The authors thank D. Markowitz, the IARPA MICrONS programme Manager, for his support during all three phases of the MICrONS programme; IARPA programme managers J. Vogelstein and D. Markowitz for co-developing the MICrONS programme; J. Wang, IARPA SETA for her assistance; and M. Bethge, M. Mathis, B. Richards, A. Zador and J. Zylberberg for many stimulating discussions regarding building foundation models for the brain. The work was supported by the MICrONS Program of Intelligence Advanced Research Projects Activity (IARPA) via Department of Interior/Interior Business Center (DoI/IBC) contract numbers D16PC00003, D16PC00004 and D16PC0005. The US Government is authorized to reproduce and distribute reprints for Governmental purposes notwithstanding any copyright annotation thereon. X.P. and A.S.T. acknowledge support from NSF NeuroNex grant 1707400. A.S.T. also acknowledges support from the National Institute of Mental Health and National Institute of Neurological Disorders And Stroke under award number U19MH114830 and National Eye Institute award numbers R01 EY026927 and Core Grant for Vision Research T32-EY-002520-37. Disclaimer: the views and conclusions contained herein are those of the authors and should not be interpreted as necessarily representing the official policies or endorsements, either expressed or implied, of IARPA, DoI/IBC, or the US Government. A.S.E. received funding from the European Research Council (ERC) under the European Union's Horizon Europe research and innovation programme (grant agreement 101041669) as well as the Deutsche Forschungsgemeinschaft (DFG, German Research Foundation), project ID 432680300 (S.F.B. 1456, project B05).

**Author contributions** We adopted the following contribution categories from CRediT (Contributor Roles Taxonomy). Authors within each category are sorted in the same order as in the author list. Conceptualization: E.Y.W. and A.S.T. Methodology: E.Y.W., F.H.S. and A.S.T. Software: E.Y.W. and F.H.S. Validation: E.Y.W., P.G.F., Zhuokon Ding, M.A.W., Zhiwei Ding, D.T. and J.F. Formal analysis: E.Y.W., S. Papadopoulos and A.C. Investigation: E.Y.W., P.G.F., K.P. and T.M. Resources: P.G.F., S. Patel, S. Papadopoulos, C.M.S.-M., R.C.R., F.C. and N.M.d.C. Data Curation: P.G.F., S. Papadopoulos. Writing, original draft: E.Y.W., P.G.F., S. Papadopoulos, F.H.S. and A.S.T. Writing, review and editing: E.Y.W., P.G.F., S. Papadopoulos, K.F., A.S.E., J.R., X.P., F.H.S. and A.S.T. Visualization: E.Y.W. and S. Papadopoulos. Supervision, project administration, and funding acquisition: A.S.T.

**Competing interests** A.S.T. and J.R. are co-founders of DataJoint Inc., in which they have financial interests. The other authors declare no competing interests.

**Additional information**
**Correspondence and requests for materials** should be addressed to Andreas S. Tolias.

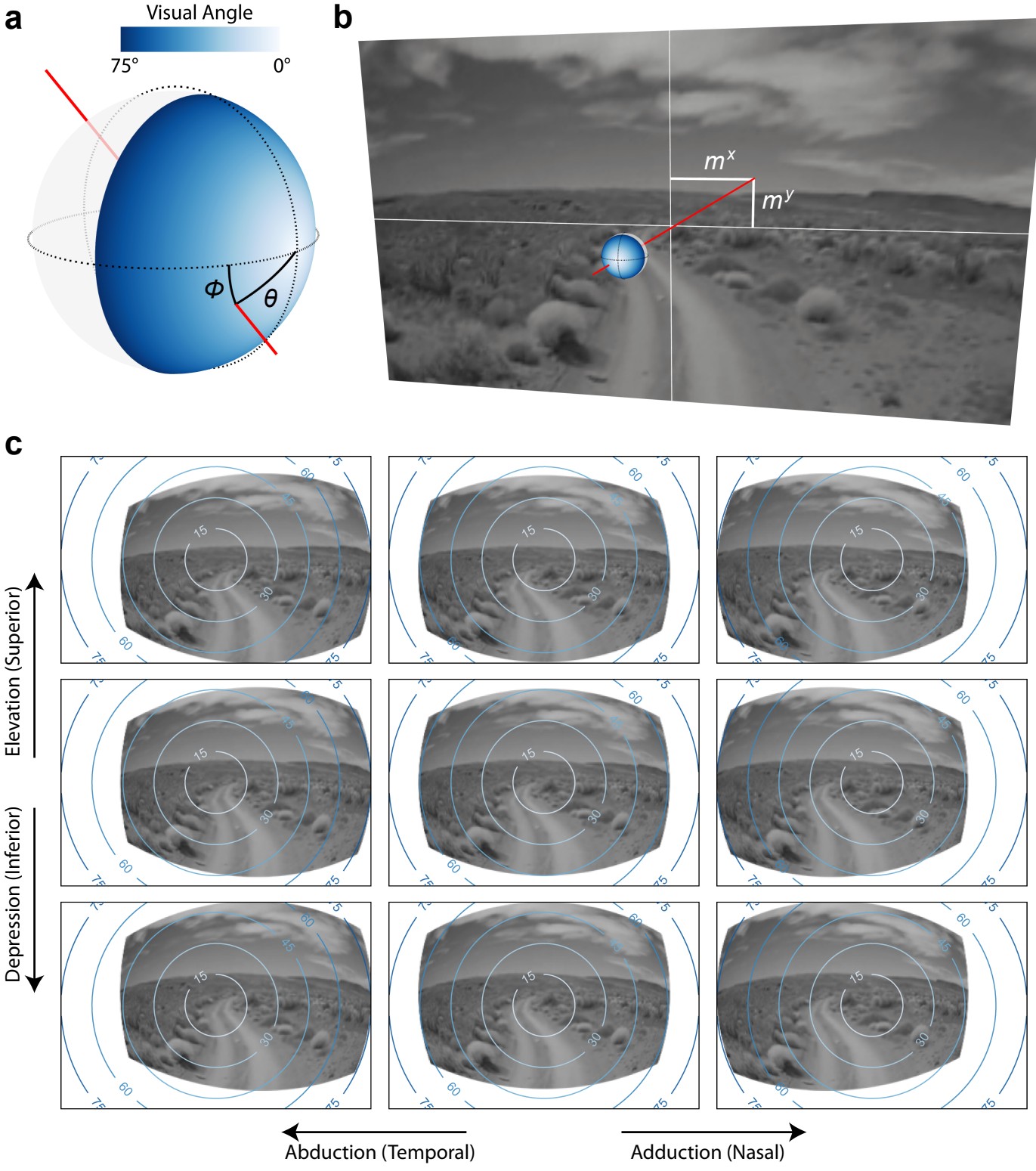

**Extended Data Fig. 1 | ANN perspective.** Schematic of the modeled perspective the animal. **a**, The retina is modeled as points on a sphere receiving light rays that trace through the origin. An example light ray with polar angle $\theta$ and azimuthal angle $\phi$ is shown in red. **b**, The light ray is traced to a point $m^x$, $m^y$ on the monitor. Bilinear interpolation of the four pixels on the monitor surrounding $m^x$, $m^y$ produces the activation of a point $\theta$, $\phi$ on the modeled retina. **c**, 9 examples of the modeled perspective from the left eye of an animal, with 3 horizontal rotations of the optical globe (abduction/adduction) × 3 vertical rotations (elevation/depression). The concentric circles indicate visual angles in degrees. (See Methods for details on the perspective network).

**a** Pupil Size (au)

Small Large

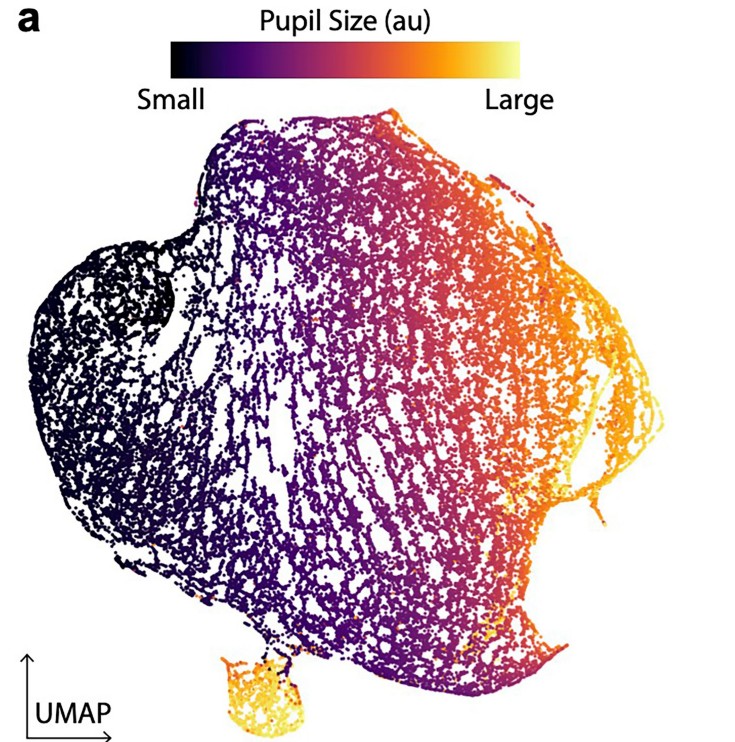

UMAP

**b** Treadmill Speed (au)

Still Fast

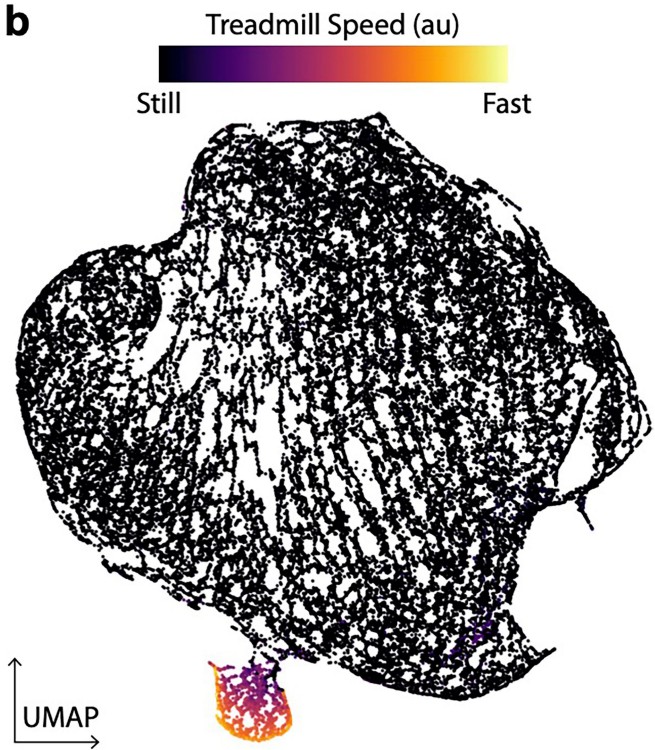

UMAP

**Extended Data Fig. 2 | ANN modulation.** Visualization of the modulation network's output, projected onto 2 dimensions via UMAP. **a**, **b** show the same data from an example recording session and modulation network. Each point on the plot indicates a point in time from the recording session. The colors indicate measurements of pupil size (**a**) and treadmill speed (**b**) at the respective points in time. (See Methods for details on the modulation network).

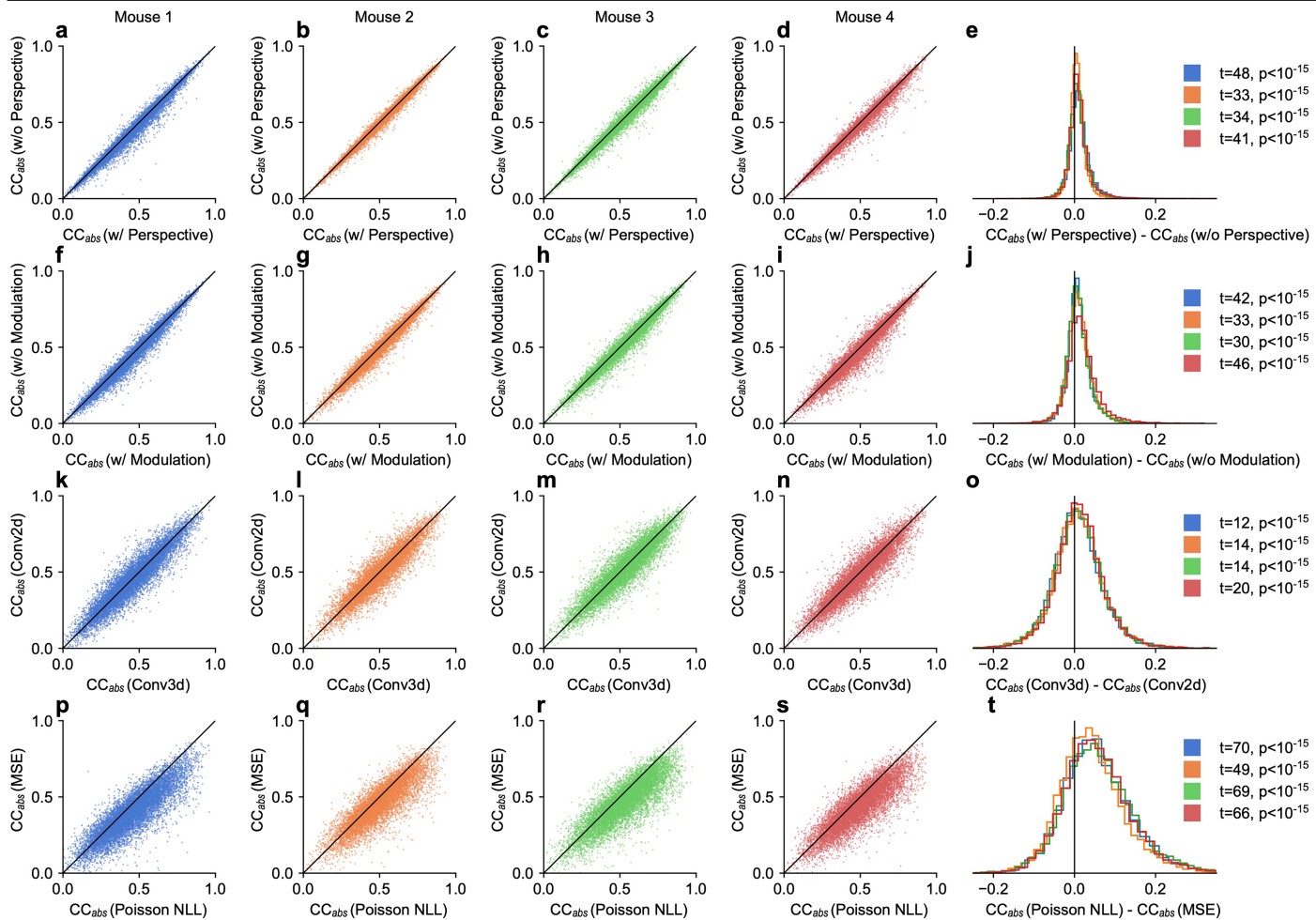

**Extended Data Fig. 3 | Neural network lesion studies.** To determine the effect that various components of the model have on predictive accuracy, we performed lesion studies, where we altered individual components of model and evaluated the effect that the alteration had on model performance ($CC_{abs}$). The left 4 columns (**a-d**, **f-i**, **k-n**, **p-s**) are scatterplots of reference vs lesioned model performance, with each column corresponding to different mouse and each point corresponding to a neuron. The right-most column (**e**, **j**, **o**, **t**) displays density histograms of the performance difference between the reference and the lesioned models, plotted separately for each mouse, as well as the t-statistic and p-values of paired two-sided t-tests. The first row (**a-e**) shows the effect of the perspective module on model performance, the second row (**f-j**) shows the effect of the modulation module, the third row (**k-o**) shows the effect of the convolution type – 2D vs 3D – of the feedforward module, and the fourth row (**p-t**) shows the effect of the loss function – Poisson negative log likelihood (Poission NLL) vs mean square error (MSE).

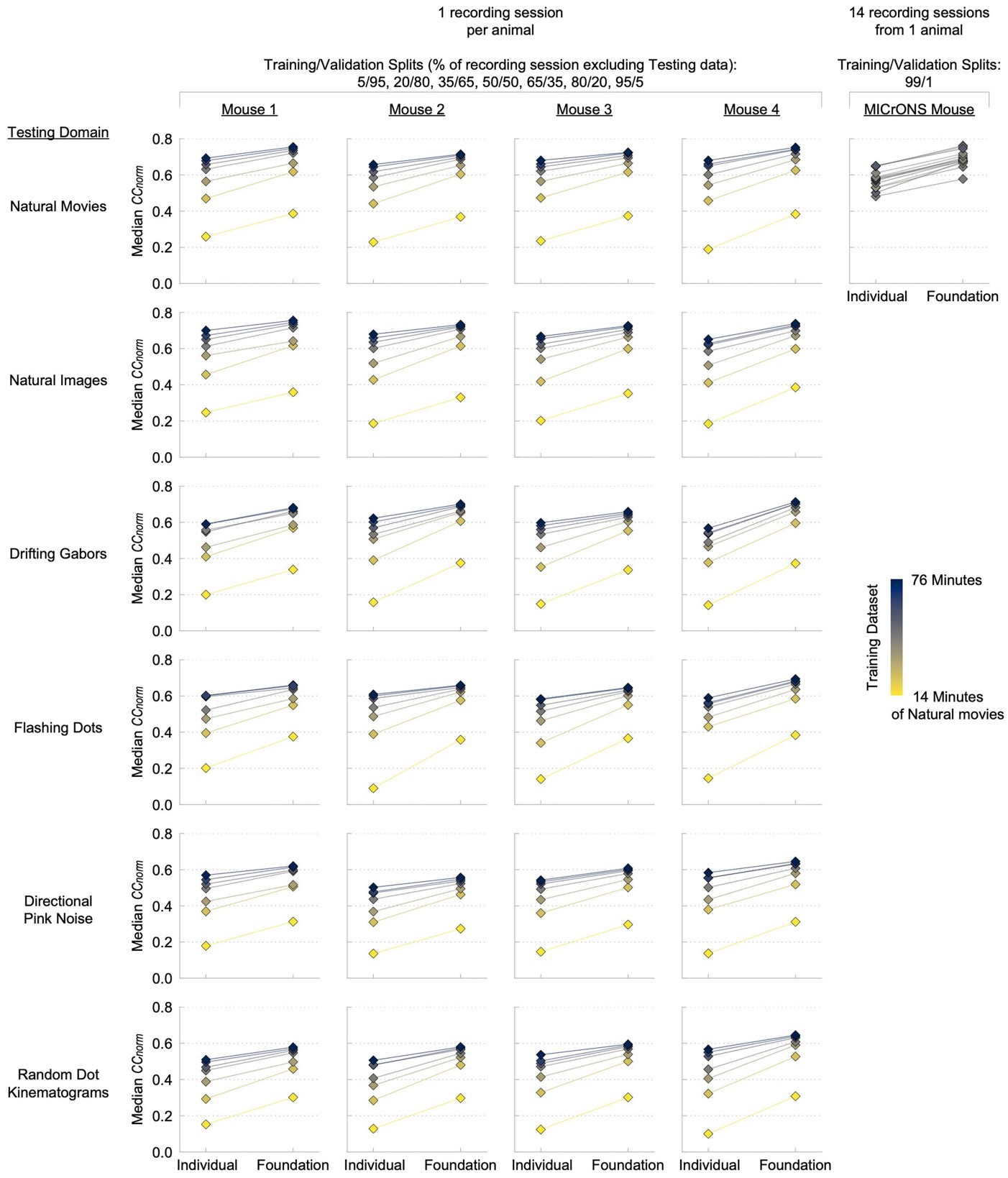

**Extended Data Fig. 4 | ANN performance: Individual vs. Foundation.** Predictive accuracy (median $CC_{norm}$ across neurons) of foundation models (with the foundation core) vs. individual models (with cores trained on individual recording sessions). For the 4 mice in the 4 left columns, 1 recording session was performed, and that data was partitioned into 7 training/validation splits, which were used to train separate individual/foundation models. The predictive accuracy of those models (diamonds) is reported for 6 testing stimulus domains (rows). For the MICrONS mouse, 14 recording sessions were performed, for each recording session, a model was trained using nearly all (99%) of the data available for training/validation. The MICrONS models were only tested on the natural movies, due to the lack of the other stimuli in the recording sessions. All models were trained only using natural movies.

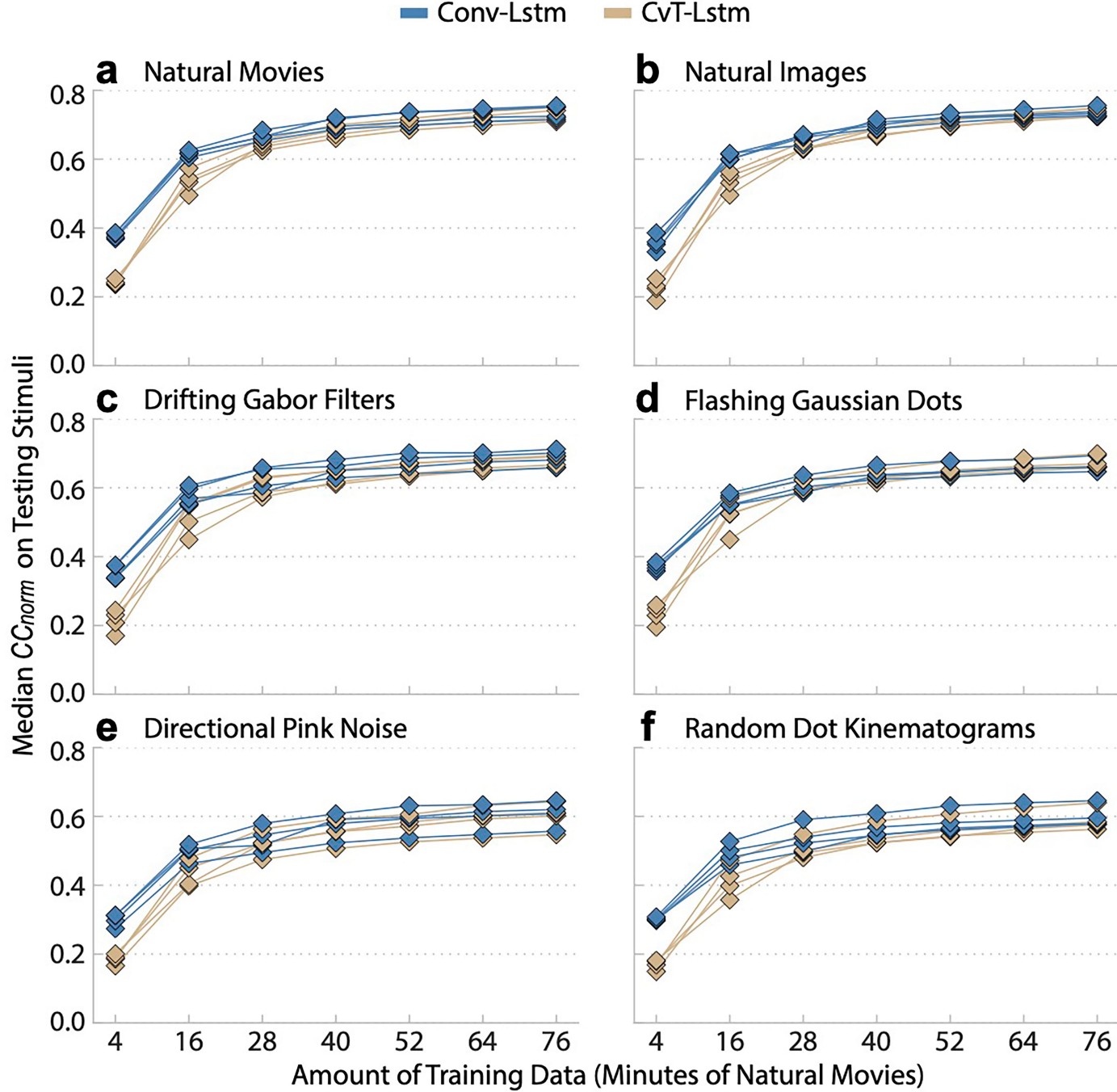

**Extended Data Fig. 5 | Recurrent architecture: Conv-Lstm vs. CvT-Lstm.**
We evaluated the performance of two different types of recurrent architectures for the core module: Conv-Lstm (blue) and CvT-Lstm (tan). For each architecture, a core was trained on 8 mice and then transferred to 4 new mice. For each of the new mice, 7 models were trained using varying amounts of natural movies, ranging from 4 to 76 minutes. The predictive accuracy ($CC_{norm}$) of these models was evaluated on 6 different stimulus domains: natural movies (**a**), natural images (**b**), drifting gabor filter (**c**), flashing Gaussian dots (**d**), directional pink noise (**e**), random dot kinematograms (**f**). Blue diamonds indicate models with the Conv-Lstm core, and tan diamonds indicate models with the CvT-Lstm core. For each architecture, models of the same mouse are connected by lines.

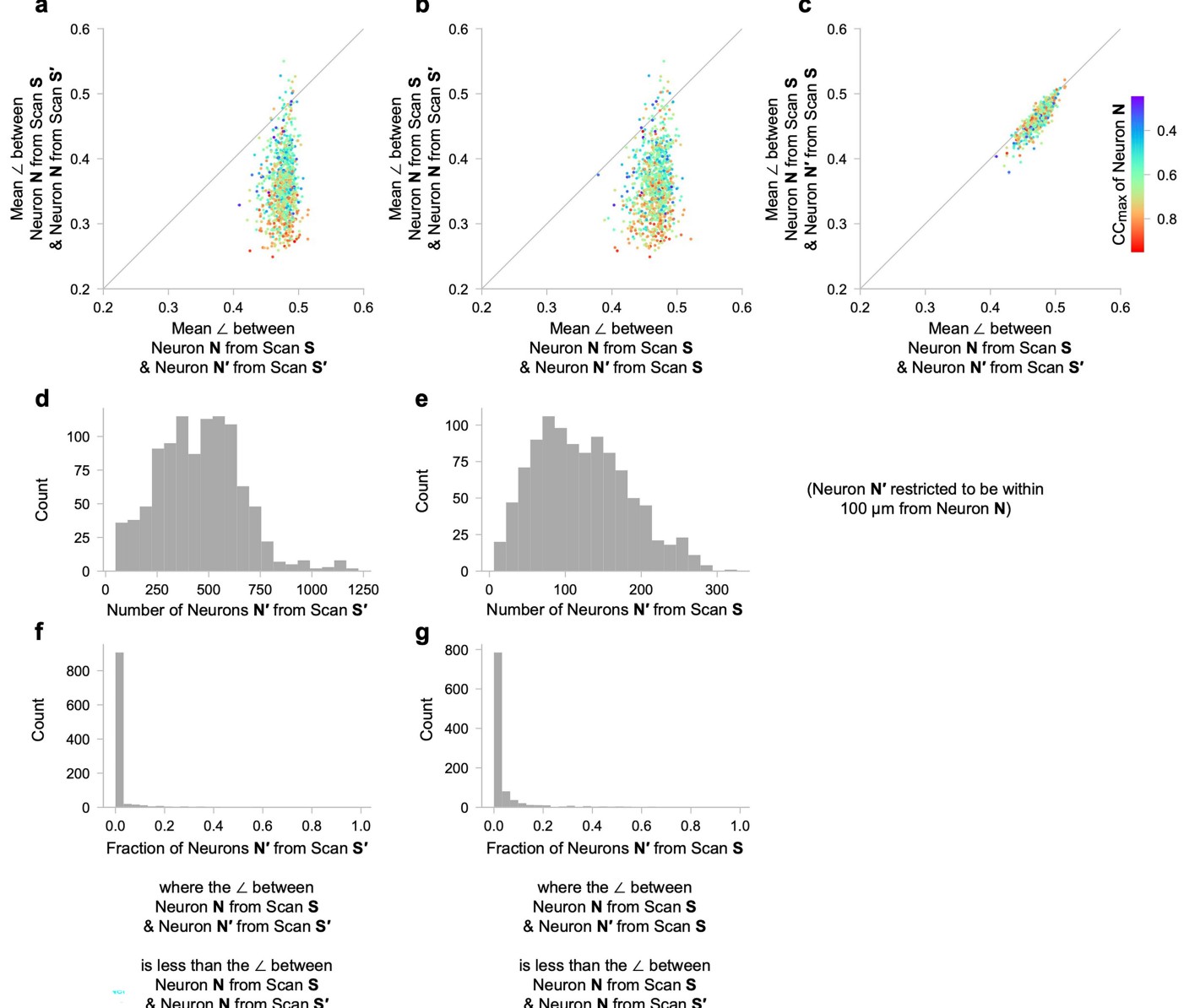

(Neuron **N'** restricted to be within
100 μm from Neuron **N**)

**Extended Data Fig. 6 | Pairwise similarities of readout feature weights of neurons from the MICrONS volume.** Here we examine the similarities of readout weights of same or different neurons, from same or different scans (recording sessions). In panels **a**–**c**, the similarities of readout weights are plotted for the following groups: *same neuron* from *different scan* (y-axis of **a**), *same neuron* from *same scan* (y-axis of **b**), *different neuron* from *different scan* (x-axis of **a**, x-axis of **c**), *different neuron* from *same scan* (x-axis of **b** and y-axis of **c**). The similarity between readout weights was measured inversely via angular distance ∠ := arccos((**x** · **y**)/(‖**x**‖‖**y**‖))/π, where **x**, **y** is a pair of readout weights. A similar pair of readout weights will exhibit a small ∠, and vice versa. The scatterplots **a**–**c** are colored by the $CC_{max}$, which is an inverse measure of

neuronal noise, i.e., the estimated maximum correlation coefficient that a model could achieve at predicting the mean response the neuron (see Methods for details). For each neuron **N**, the 'different' neuron **N'** was restricted to be ≤100 μm apart from each other in terms of soma distance, and the distribution of the number of 'different' neurons is shown in **d** (from different scans) and **e** (from the same scan). **f** and **g** (corresponding to **d** and **e**, respectively) show the fraction of the nearby neurons **N'** that are more similar to **N** in terms of readout weights than **N** is to itself across different scans. **f**, For 919 out of the 1013 neurons **N**, less than 0.05 of nearby neurons **N'** from different scans had more similar readout weights. **g**, For 840 out of the 1013 neurons **N**, less than 0.05 of nearby neurons **N'** from the same scan had more similar readout weights.

**Extended Data Table 1 | Table listing the experimental recordings, collected for either foundation core training (Foundation Cohort = Yes) or validation (Foundation Cohort = No)**

| Foundation Cohort | Figure | Animal ID | Neurons | Visual Areas | Training Data — Natural Movies | Testing Data — Natural Movies | Natural Images | Drifting Gabor Filters | Flashing Gaussian Dots | Directional Pink Noise | Random Dot Kinematogram |
|---|---|---|---|---|---|---|---|---|---|---|---|
| Yes | 3a | 25133-12-14 | 10328 | V1, LM, AL, RL | 24x1 + 24x2 + 15x4 | 1x20 | | | | 5x4 | |
| Yes | 3a | 25312-2-24 | 7508 | V1, LM, AL, RL | 24x1 + 24x2 + 15x4 | 1x20 | | | | 5x4 | |
| Yes | 3a | 25404-4-20 | 9346 | V1, LM, AL, RL | 24x1 + 24x2 + 15x4 | 1x20 | | | | 5x4 | |
| Yes | 3a | 25505-3-11 | 9346 | V1, LM, AL, RL | 24x1 + 24x2 + 15x4 | 1x20 | | | | 5x4 | |
| Yes | 3a | 24620-9-13 | 7715 | V1, LM, AL, RL, AM, PM | 96x1 | 1x15 | | | | | |
| Yes | 3a | 25702-5-16 | 7114 | V1, LM, AL, RL, AM, PM | 96x1 | 1x20 | | | | | |
| Yes | 3a | 25830-3-9 | 6445 | V1, LM, AL, RL, AM, PM | 96x1 | 1x20 | | | | | |
| Yes | 3a | 25833-3-13 | 7805 | V1, LM, AL, RL, AM, PM | 96x1 | 1x20 | | | | | |
| No | 2b-c, 3a-g | 26872-19-13 | 10728 | V1, LM, AL, RL | 80x1 | 1x10 | 1x10 | 1x10 | 1x10 | 1x10 | 1x10 |
| No | 2b-c, 3a-g | 27203-4-7 | 8429 | V1, LM, AL, RL | 80x1 | 1x10 | 1x10 | 1x10 | 1x10 | 1x10 | 1x10 |
| No | 2b-c, 3a-g | 27204-3-13 | 10126 | V1, LM, AL, RL | 80x1 | 1x10 | 1x10 | 1x10 | 1x10 | 1x10 | 1x10 |
| No | 2b-c, 3a-g | 27342-4-12 | 9478 | V1, LM, AL, RL | 80x1 | 1x10 | 1x10 | 1x10 | 1x10 | 1x10 | 1x10 |
| No | 4a-i | 27204-4-8 | 10336 | V1, LM, AL, RL | 60x1 | 1x10 | | | 30x1 | 30x1 | |
| No | 4a-i | 27424-4-13 | 9614 | V1, LM, AL, RL | 60x1 | 1x10 | | | 30x1 | 30x1 | |
| No | 4a-i | 27468-4-17 | 10454 | V1, LM, AL, RL | 60x1 | 1x10 | | | 30x1 | 30x1 | |

The animal ID, number of neurons, and areas of the visual cortex are listed for each experiment. The 'Training Data' and 'Testing Data' columns list the Minutes x Repeats of each type of stimulus, designated for either model training or testing.

# Reporting Summary

## Statistics

For all statistical analyses, confirm that the following items are present in the figure legend, table legend, main text, or Methods section.

| n/a | Confirmed | |
|---|---|---|
| ☐ | ☒ | The exact sample size (*n*) for each experimental group/condition, given as a discrete number and unit of measurement |
| ☐ | ☒ | A statement on whether measurements were taken from distinct samples or whether the same sample was measured repeatedly |
| ☐ | ☒ | The statistical test(s) used AND whether they are one- or two-sided *Only common tests should be described solely by name; describe more complex techniques in the Methods section.* |
| ☒ | ☐ | A description of all covariates tested |
| ☒ | ☐ | A description of any assumptions or corrections, such as tests of normality and adjustment for multiple comparisons |
| ☐ | ☒ | A full description of the statistical parameters including central tendency (e.g. means) or other basic estimates (e.g. regression coefficient) AND variation (e.g. standard deviation) or associated estimates of uncertainty (e.g. confidence intervals) |
| ☐ | ☒ | For null hypothesis testing, the test statistic (e.g. *F*, *t*, *r*) with confidence intervals, effect sizes, degrees of freedom and *P* value noted *Give P values as exact values whenever suitable.* |
| ☒ | ☐ | For Bayesian analysis, information on the choice of priors and Markov chain Monte Carlo settings |
| ☒ | ☐ | For hierarchical and complex designs, identification of the appropriate level for tests and full reporting of outcomes |
| ☒ | ☐ | Estimates of effect sizes (e.g. Cohen's *d*, Pearson's *r*), indicating how they were calculated |

*Our web collection on statistics for biologists contains articles on many of the points above.*

## Software and code

Policy information about availability of computer code

| | |
|---|---|
| Data collection | For image acquisition, we used ScanImage 2017b. Stimuli were presented using PsychToolBox 3. The data collection process was automated with Labview. |
| Data analysis | We used DeepLabCut (2.0.5) for automatic tracking of the pupil. We used CaImAn (1.0) for automatic segmentation and deconvolution of calcium imaging data. Our custom built analysis pipeline (https://github.com/cajal/pipeline, https://github.com/cajal/foundation) also used general tools like Numpy (1.23.5), pandas (1.5.3), SciPy (1.10.1), statsmodels (0.13.5), scikit-learn (1.2.1), PyTorch (1.12.1), Matplotlib (3.7.0), seaborn (0.12.2), HoloViews (1.15.4), Ipyvolume (0.5.2), Jupyter (ipykernel: 6.21.2), MySQL (5.7.37), Docker (23.0.1), and Kubernetes (1.22.11). DataJoint (0.12.9) were used for storing and managing data. |

For manuscripts utilizing custom algorithms or software that are central to the research but not yet described in published literature, software must be made available to editors and reviewers. We strongly encourage code deposition in a community repository (e.g. GitHub). See the Nature Portfolio guidelines for submitting code & software for further information.

## Data

Policy information about availability of data

All manuscripts must include a data availability statement. This statement should provide the following information, where applicable:

- Accession codes, unique identifiers, or web links for publicly available datasets
- A description of any restrictions on data availability
- For clinical datasets or third party data, please ensure that the statement adheres to our policy

> The MICrONS functional and structural data are available on BossDB (https://bossdb.org/project/microns-minnie, please also see https://www.microns-explorer.org/cortical-mm3 for details). The MICrONS foundation model is available on GitHub (https://github.com/cajal/fnn).

## Research involving human participants, their data, or biological material

Policy information about studies with human participants or human data. See also policy information about sex, gender (identity/presentation), and sexual orientation and race, ethnicity and racism.

| | |
|---|---|
| Reporting on sex and gender | *Use the terms sex (biological attribute) and gender (shaped by social and cultural circumstances) carefully in order to avoid confusing both terms. Indicate if findings apply to only one sex or gender; describe whether sex and gender were considered in study design; whether sex and/or gender was determined based on self-reporting or assigned and methods used. Provide in the source data disaggregated sex and gender data, where this information has been collected, and if consent has been obtained for sharing of individual-level data; provide overall numbers in this Reporting Summary. Please state if this information has not been collected. Report sex- and gender-based analyses where performed, justify reasons for lack of sex- and gender-based analysis.* |
| Reporting on race, ethnicity, or other socially relevant groupings | *Please specify the socially constructed or socially relevant categorization variable(s) used in your manuscript and explain why they were used. Please note that such variables should not be used as proxies for other socially constructed/relevant variables (for example, race or ethnicity should not be used as a proxy for socioeconomic status). Provide clear definitions of the relevant terms used, how they were provided (by the participants/respondents, the researchers, or third parties), and the method(s) used to classify people into the different categories (e.g. self-report, census or administrative data, social media data, etc.) Please provide details about how you controlled for confounding variables in your analyses.* |
| Population characteristics | *Describe the covariate-relevant population characteristics of the human research participants (e.g. age, genotypic information, past and current diagnosis and treatment categories). If you filled out the behavioural & social sciences study design questions and have nothing to add here, write "See above."* |
| Recruitment | *Describe how participants were recruited. Outline any potential self-selection bias or other biases that may be present and how these are likely to impact results.* |
| Ethics oversight | *Identify the organization(s) that approved the study protocol.* |

Note that full information on the approval of the study protocol must also be provided in the manuscript.

# Field-specific reporting

Please select the one below that is the best fit for your research. If you are not sure, read the appropriate sections before making your selection.

☒ Life sciences ☐ Behavioural & social sciences ☐ Ecological, evolutionary & environmental sciences

For a reference copy of the document with all sections, see nature.com/documents/nr-reporting-summary-flat.pdf

# Life sciences study design

All studies must disclose on these points even when the disclosure is negative.

| | |
|---|---|
| Sample size | No sample-size calculation was performed a priori. Sample sizes (number of connections tested) match or exceed previous studies of similar design. |
| Data exclusions | Of the 14 released MICrONS scans, one scan was excluded a priori from the study due to experimental issues (responses to some stimuli were not collected due to water running out from the objective). Duplicate detection was performed to identify neurons that were recorded more than once in experiments. Besides that, no neurons were exlcuded from model training or analysis. |
| Replication | The approach of using a foundation model core to fit new models of mice was replicated across 4 mice for evaluating predictive accuracy (Figure 3) and 3 mice for evaluating parametric tuning accuracy (Figure 4). |
| Randomization | No randomization of animal subjects was performed as our experimental design did not stratify into animal groups. |
| Blinding | No blinding is performed during data collection since our study did not include predefined experimental groups for sample allocation. The |

| Blinding | analysis is performed unblinded; however, the same computational methods were applied to all control and sample groups. |

# Reporting for specific materials, systems and methods

We require information from authors about some types of materials, experimental systems and methods used in many studies. Here, indicate whether each material, system or method listed is relevant to your study. If you are not sure if a list item applies to your research, read the appropriate section before selecting a response.

## Materials & experimental systems

| n/a | Involved in the study |
|---|---|
| ☒ | Antibodies |
| ☒ | Eukaryotic cell lines |
| ☒ | Palaeontology and archaeology |
| ☐ ☒ | Animals and other organisms |
| ☒ | Clinical data |
| ☒ | Dual use research of concern |
| ☒ | Plants |

## Methods

| n/a | Involved in the study |
|---|---|
| ☒ | ChIP-seq |
| ☒ | Flow cytometry |
| ☒ | MRI-based neuroimaging |

## Animals and other research organisms

Policy information about studies involving animals; ARRIVE guidelines recommended for reporting animal research, and Sex and Gender in Research

| Laboratory animals | For experiments excluding the MICrONS dataset in this manuscript: Three mice, Mus musculus, 78-86 days old at first experimental scan. Heterozygous for both Slc17a7-Cre (B6;129S-Slc17a7tm1.1(cre)Hze/J, Jackson Laboratory Strain # 023527) and Ai162 (B6.Cg-Igs7tm162.1(tetO-GCaMP6s,CAG-tTA2)Hze/J, Jackson Laboratory Strain # 031562). The MICrONS dataset was collected from a mouse of the same species and strain, 75 days old. |
|---|---|
| Wild animals | Study did not involve wild animals. |
| Reporting on sex | For new experiments in this manuscript: 6 Female, 8 Males. For MICrONS dataset, 1 Male. Animals were randomly recruited to the study with respect to sex. Analysis disaggregated for sex was not performed, due to low sample size and expected generalization of principles under study across genders. |
| Field-collected samples | Study did not involve samples collected from the field. |
| Ethics oversight | All procedures were approved by the Institutional Animal Care and Use Committee of Baylor College of Medicine. |

Note that full information on the approval of the study protocol must also be provided in the manuscript.

## Plants

| Seed stocks | *Report on the source of all seed stocks or other plant material used. If applicable, state the seed stock centre and catalogue number. If plant specimens were collected from the field, describe the collection location, date and sampling procedures.* |
|---|---|
| Novel plant genotypes | *Describe the methods by which all novel plant genotypes were produced. This includes those generated by transgenic approaches, gene editing, chemical/radiation-based mutagenesis and hybridization. For transgenic lines, describe the transformation method, the number of independent lines analyzed and the generation upon which experiments were performed. For gene-edited lines, describe the editor used, the endogenous sequence targeted for editing, the targeting guide RNA sequence (if applicable) and how the editor was applied.* |
| Authentication | *Describe any authentication procedures for each seed stock used or novel genotype generated. Describe any experiments used to assess the effect of a mutation and, where applicable, how potential secondary effects (e.g. second site T-DNA insertions, mosiacism, off-target gene editing) were examined.* |

