## [Peer Review File · Nature]

Foundation model of neural activity predicts response to new stimulus types and anatomy

Corresponding Author: Professor Andreas Tolias

Version 0:

Reviewer comments:

Referee #1

(Remarks to the Author)

The authors show that an artificial neural network (ANN) can be trained to reproduce recorded calcium imaging data for over 70k neurons in the visual cortex of the mouse. Furthermore, the ANN can predict very well the neural responses to visual stimuli that were not included in the training set. Substantial improvements over similar results from the same and other labs were achieved by including data on the current perspective of the mouse, inferred from eye-tracking data, and its behavioral state, inferred from locomotion and pupil dilation.

These results are certainly very nice and useful. But I do not find them so significant that they merit a publication in Nature. The insight which they provide into the structure or function of the visual cortex is limited. For example, it remains unknown how the features which the ANN extracts from the visual stimuli and the other recorded data are related to features that are extracted by the visual cortex. The authors make no effort to advance our insight on that. One underlying problem is that the recorded response of each neuron is predicted by a readout, a trained weighted sum of large numbers of features that the ANN extracts. These readout weights are adapted individually for each neuron. One cannot expect that these weighted sums have any relation to the synaptic input which the neuron receives in the brain, especially if one uses readouts from dozens or more units of the ANN. Hence in a sense, one just replaces one black box, the visual cortex, by another black box, the trained ANN. But since the trained ANN provides a new representation of the recorded data, it would be of interest to see whether they actually provide a compression of the recorded data. But this is not pursued in this study.

I was missing quantitative data on the architecture of the ANN model; I did not even find how many parameters are trained in a readout for a specific neuron. One could also debate whether the term "foundation model" is adequate for this trained ANN. Most currently discussed foundation models such as GPT or BERT, are based on transformers rather than standard ANNs. Apparently it remains unknown whether these newer types of models can provide further improvements for predicting large-scale neural recordings from the brain.

Referee #2

(Remarks to the Author)

In this paper, Wang et al. present a deep neural network model trained to predict the responses of neurons in mouse visual cortex to natural movies. The model consists of four components: a perspective network for transforming images into retinal activations, a modulation network for conditioning the responses on the animal's behavior, a core network for transforming retinal activations into a latent space via 3D convolutions and recurrence, and a readout network for transforming the latent representation into neural activations. The authors show that they can train on data from multiple animals using the same core network, which then comprises something the authors present as being akin to a "foundation model" for predicting neural responses, where the term "foundation model" as used in the field of AI indicates a pre-trained backbone that can be fine-tuned on new downstream tasks. Here, it is shown that if they pre-train the foundation core with multiple animals then train on new animals' data they can achieve better correlations between predicted activity and real activity than they achieve by training the model on each individual animal, particularly for low amounts of training data. They also show that the model can generalize to new stimuli and new types of stimuli, e.g. artificial stimuli rather than natural movies. The authors go on to show that thanks to the improved response predictions the model can be used to estimate basic properties of each neuron, including orientation selectivity, direction selectivity, spatial selectivity, and retinotopic position, which they verify using data

from the MICrONS dataset.

In general, this paper is well-written and the authors have achieved an impressive technical feat. This model is achieving something that past models could not, namely, accurate prediction of responses to movies and generalization to new stimulus types. That is commendable and a clear contribution. However, evidence that this is really a novel approach or that it constitutes a “foundation model” is actually quite limited, and the demonstrations provided of the model’s utility are a little bit underwhelming given how impressive the model is otherwise. As well, given that this paper ultimately presents a methodological advance, i.e. no novel information is learned about mouse visual cortex, it is important to provide more technical information on the development of the model than is provided currently. Below, I expand on these points, which I believe would need to be addressed to make this paper appropriate for publication in Nature. I then provide a few minor comments/questions.

Major concerns

1) What is the novel approach here, and is this really a “foundation model”?

As noted in the paper, other papers (e.g. Lurz et al. (2020)) previously showed a transfer learning effect from training on one group of animals and then fine-tuning on another, so the major jump here seems to be using 3D convolutions and recurrence to make predictions on movies rather than still images. That is not a trivial engineering achievement, but is it really that novel an approach? In line with this concern, the implication is that this work is novel because it presents a “foundation model”. But, one of the most interesting aspects of traditional foundation models in AI, like large language models, is that they not only generalize to new data but also to new *tasks*. In contrast, here, the model is being transferred to the same task (predict neural responses from movies), and it is simply generalizing to new data in that same task. Could the authors provide an example of generalization to a novel task? Without this, it is arguably not a “foundation model”.

For example, could the model be fine-tuned to predict the sub-type of cells (e.g. pyramidal neuron, Martinotti cell, etc.) or which cells are synaptically connected to one another? These things could be done by taking the foundation core and the existing read-out layer for activity then adding a new secondary classification read-out layer that receives the predicted activity (and also potentially the latent representations) and outputs cell type or a set of connection probabilities. This could then be fine-tuned on the structural information from the MICrONS dataset, which bizarrely enough, is used only for its recordings and positional information and not its rich structural information, despite the hint at the beginning of the paper of this possibility. In fact, as far as I can tell, nothing is done to actually use the EM component of the MICrONS data here, which is an odd choice given that it is highlighted in the abstract and Figure 5. Hence, the downstream task of predicting cell types or synaptic connections seems like an obvious choice as a novel task to fine-tune on. But, if the authors can think of another task that is fine. The important thing is simply to show generalization to a novel domain other than predicting neural responses. Without that this isn’t a “foundation model”, but rather, just a pre-trained model being used for transfer learning, as has been done before.

2) What could we actually use a model like this for other than reducing the recording time required in experiments? What could we learn?

Though the engineering feat they have engaged in is impressive, there is relatively little done with it in the way of learning something new. What is shown in this paper is that the model can be used to predict simple tuning curve properties that can easily be measured with more recording time, so the model could theoretically save on recording time in experiments. But, is that really a major advance worthy of publication in Nature? Arguably, the authors should at least provide some speculation about what else we could actually learn from such a model, and if possible, even demonstrate or confirm one piece of such speculation. Could we predict previously unknown functional cell types? Could we predict learning trajectories to verify theories of learning rules? What can be done other than simply saving time in experiments? This is important for demonstrating that this work is something more than a simple extension of previous models for fitting neural responses to the temporal domain. Even some speculation here would help strengthen the paper.

3) How were the design choices for the network arrived, e.g. how were hyperparameter and architecture choices made?

Given that this is largely a methodological paper, one critical piece missing is a more thorough description and analysis of the design choices made in the model and their impact. For example, some “lesion” studies should be included on the different components of the model. How important is the perspective or modulation network for performance? What if the loss function at the output is changed? What if 2D convolutions are used instead of 3D convolutions? Similarly, how were hyperparameter choices made (learning rate schedule, batch size, momentum, etc.)? Why was the particular optimizer that was used selected? And so on...

In the absence of these kind of details it will be more difficult for other scientists to learn from the experience here and expand on this model to build new models for similar purposes. Hence, these sorts of matters are very important for a paper like this.

Minor comments

* Why have the different “layers” in the recurrent nets in the core? If they’re all connected to each other bidirectionally, is this not just equivalent to having one recurrent network with three times the number of channels? Is there something I’m missing?

* Does Figure 5b do anything? The EM data is not used here at all as far as I can tell. It is weird to include an image of it and then do nothing with it.

* For the OSI and DSI estimates, how does the reliability of the model compare to the reliability one would get from multiple experiments in the same animal?

* It took me a while to understand what was being shown in Extended Data Figure 4a-c. I think there might be a way to make it easier to understand, e.g., by simplifying the axes labels or providing a bit more explanation of what the data shows in the caption.

Version 1:

Reviewer comments:

Referee #2

(Remarks to the Author)

The authors have done an excellent job of addressing my concerns. Specifically, the new data provided in Figure 5 and Extended Figure 3 attend to my most pressing concerns.

My only final recommendation would be that I think it is a bit odd to finish the Results section with references to results in other papers (which is what’s done in the final paragraph of the revised manuscript). I think that Figure 5 is now enough to make the point that this is really pushing towards a proper foundation model, and these other studies can simply be mentioned in the discussion.

(Remarks on code availability)

The README file should contain useful information, not just a title.

The images or other third party material in this Peer Review File are included in the article’s Creative Commons license, unless indicated otherwise in a credit line to the material. If material is not included in the article’s Creative Commons license and your intended use is not permitted by statutory regulation or exceeds the permitted use, you will need to obtain permission directly from the copyright holder.

General response

We thank the reviewers for their thoughtful comments. We were pleased that they found our paper “well written” and that our model “achieved an impressive technical feat” accomplishing “something that past models could not” in terms of “accurate prediction of responses to movies and generalization to new stimulus types”. They also expressed several constructive criticisms and made interesting suggestions. Specifically, they suggested that we should demonstrate its utility not only to generalize to new stimulus domains outside its training distribution but also to new tasks that are not predicting neural activity to strengthen our claim that this constitutes a foundation model. They also wanted more information regarding the utility of our model to extract novel insights about the visual cortex and how it can go beyond replacing “one black box, the visual system, with another black box.” The reviewers also requested a comprehensive lesion analysis to demonstrate the contribution of the different components of model architecture and comparison to transformer networks, which are now popular in AI research. We’ve addressed all reviewers’ concerns, highlighting our model’s novel neuroscience discoveries. These include uncovering nuanced synaptic connectivity rules and linking functional properties to unknown structural features. By demonstrating our model’s ability to predict structure, not just neural activity, we’ve strengthened our claim that it constitutes a brain foundation model. Additionally, we have addressed other more minor issues by conducting additional experiments, analyses, and developing new models, which we believe have substantially improved the manuscript. Given that the main criticisms from the reviewers largely overlap, we first discuss them below and then provide a point-by-point response to each comment.

Beyond Neural Prediction and foundation models of the brain.

In AI research, foundation models are defined as a class of models that are pre-trained on large amounts of data from many sources, capable of performing a variety of downstream tasks they were not originally trained for with no or minimal fine-tuning. We believe that our model is one important step towards building such a model for neuroscience. The point of discussion is to what degree it generalizes to new situations. In the previous version of our manuscript we had already demonstrated the following:

- The model generalizes to new neurons and animals with minimal training.
- The model generalizes to new stimulus domains (e.g., classical parametric stimuli like coherent random dots, dynamic noise patterns) that have vastly different statistics from the training set. In machine learning, a comparable example would be a model generalizing from natural videos to drawings, cartoons, or rendered scenes (i.e., out-of-distribution generalization, which is a challenging problem for AI).
- Based on our model’s ability to generalize to parametric stimuli it was not trained on, it also accurately predicted the classic tuning properties of single neurons.

It is true that Lurz et al. [1] have already demonstrated that models trained on several scans exhibit good transfer learning to new neurons and animals. However, their work used static images—as opposed to video—and did not demonstrate generalization to new stimulus statistics (i.e., out-of-distribution generalization). A key contribution of our work is the ability of our model to generalize to new stimulus domains outside the training distribution, a challenge that has been particularly difficult for previous machine learning models [2]. The ability to generalize to new stimulus distributions is a fundamental property that enhances our model’s usefulness for neuroscientific research, enabling novel insights as we demonstrate in our work (see the section below “Beyond a Black Box Model”).

However, we agree with the reviewers’ opinion that demonstrating our model’s usefulness in making predictions beyond neural predictions—for example, in tasks related to anatomy and connectivity in neuroscience—would greatly strengthen its utility as a foundation brain model and as a resource for the community, and particularly relevant for the MICrONS functional connectomics data. Following these suggestions, we have conducted additional analyses and now demonstrate that our model not only generalizes to new stimulus distributions but also to new tasks. Specifically, using the foundational core of our model, we built a functional digital twin of

the MICrONS mouse functional connectomics dataset. No anatomical information from the EM data was used to build this model. The functional digital twin enabled us to extract a functional barcode—a vector embedding that describes the input-output function of each neuron. Using these functional barcodes, we now demonstrate:

- Our model can predict the neuronal cell types defined by morphological and other EM characteristics identified by [3] (Figure 5).
- In a companion paper [4] our model was used to predict a novel dendritic feature of layer 4 excitatory neurons.
- In other companion papers [5, 6, 7], our model enabled analysis of the relationships between function and synaptic-level wiring, revealing connectivity principles (see below).

In summary, our model generalizes to new neurons, animals, stimulus distributions, and tasks beyond neural prediction - justifying its status as a foundation model, as Reviewer 2 noted. This work marks the beginning of a transformative journey. As neuroscience amasses large, multi-modal datasets under natural stimuli and ethological behaviors, we anticipate a surge in efforts to build powerful brain foundation models. We've expanded our discussion to highlight the potential impact of these models on both basic and applied research.

Beyond a black box model: Example of novel insights

Our foundation model was key to creating a functional digital twin of the MICrONS mouse dataset [5]. This digital twin not only predicted morphologically defined cell types in layers 2-5 (Figure 5) but also powered four additional MICrONS studies. These investigations unveiled crucial insights into visual system circuit organization, bridging the long-standing gap between structure and function in cortical research. Our model was instrumental in enabling these breakthroughs. Three of these papers are in bioRxiv [5, 4, 6], and the fourth is being revised to include this new analysis [7]. Recognizing the valuable time of our reviewers, we have summarized these findings below to facilitate a quicker and more efficient evaluation of our foundation model's contribution to understanding brain function.

- **Functional barcodes predict connectivity [5]** Our model decomposes each neuron's visual tuning into feature (what) and spatial (where) components. The feature component is represented by a "functional barcode", which are the readout weights that map the final layer of the ANN core onto the response of the neuron. This functional barcode approach provides a more comprehensive description of neuronal function than classical methods like orientation tuning functions. Leveraging this model, Ding et al. [5] made a crucial discovery: the feature component, but not the spatial component, predicts fine-scale synaptic connectivity. This finding represents a significant advance over previous studies that relied on signal correlation to explore function-connectivity relationships. Our approach distinctly separates feature selectivity and receptive field location from underlying visual scene statistics, offering unique insight into how specific neuronal characteristics relate to connectivity.
- **Functional barcodes predict basal dendritic bias of layer 4 excitatory neurons [4]** In addition to our work presented here showing that our functional barcodes predict anatomically defined cell types, our model was also utilized by Weis et al. [4] to investigate whether a unique morphological variation of neurons in layer 4 they discovered in the MICrONS data set correlates with function. In this study they discovered that a subset of excitatory neurons at the base of layer 4 restrict their dendritic reach within layer 4, avoiding layer 5, which they quantified using a basal dendritic bias metric. Interestingly, our functional barcodes predicted this basal dendritic bias, i.e., whether a neuron avoids extending its dendrites into layer 5. The ability of our model to predict this newly discovered basal dendritic bias is significant for several reasons. Historically, neuroscience has divided the study of neurons into two main categories: functional properties associated with physiological cell types, and morphological or molecular traits linked to specific cell types. A model that establishes connections between these aspects is crucial, as it allows us to determine if and what specific structural features of neurons are related to their functions, and to explore the extent and nature of the relationship between anatomy and physiology within the complex architecture of cortical circuits. We believe our

foundation model and the functional digital twin of the MICrONS mouse marks a significant step forward in this critical area of research. Importantly when we release the MICrONS digital twin together the data already publicly available will enable the community to explore this unique functional-connectomics data and thus our model will be an invaluable resource to the community.

- **Contextual modulation in the visual cortex [6]**

Fu and collaborators [6] used deep learning models and optimized image synthesis methods to uncover novel insights into center-surround interactions in visual processing. They found that facilitating surrounds complete patterns aligned with natural image statistics, while disruptive surrounds are suppressive, consistent with Bayesian inference models of perception. These findings challenge previous results from studies using simple stimuli like gratings. To investigate how neural wiring relates to this functional principle, they then leveraged our functional digital twin of the MICrONS mouse dataset. Our model revealed that neurons with similar feature selectivity form excitatory connections regardless of receptive field overlap, a wiring pattern consistent with the observed pattern completion phenomenon. This study exemplifies how our foundation model bridges the gap between newly discovered functional principles and underlying circuit structure, enabling deeper insights into cortical information processing.

- **Hierarchical synaptic level organization in mouse V1 Layer 2/3 [7] (being revised)**

Ding and collaborators developed a novel deep learning-based method to measure single neuron invariances—the set of diverse images that strongly activate a neuron. This approach overcomes limitations of classical methods, which were mostly restricted to parametric stimuli describing for example phase invariance in V1 complex cells. Leveraging our MICrONS digital twin, they mapped these invariances of excitatory neurons in layer 2/3 of V1 and found that neurons with lower invariances were more likely to synapse onto neurons with higher invariances than onto those with lower invariances. This finding provides direct evidence for the Hubel and Wiesel hierarchy of simple-to-complex cells, a concept proposed decades ago but lacking empirical support at synaptic precision. Their analysis not only confirms this long-standing hypothesis but also demonstrates that this hierarchy exists within individual layers of V1, operating at the level of synaptic level selectivity. This study showcases how our digital twin, combined with innovative analytical approaches, can reveal fundamental principles of cortical organization that have eluded researchers for decades. (Although this result is not yet uploaded on bioRxiv as of the date we submitted our paper, we are happy to send it to the reviewers, and it should be uploaded very soon).

In summary, the results presented in our paper demonstrate the power of our digital twin model for neuroscience research. These include the ability to predict anatomically defined cell types based on functional barcodes, as well as findings from companion papers, such as predicting neuronal morphological features and relating neuronal functions to wiring. Together, these results underscore the potential of foundation models in neuroscience research. The release of the MICrONS digital twin with publication, coupled with the already public dataset (MICrONS), will provide the neuroscience community with an invaluable resource to explore this unique functional-connectomics data.

Lesion studies and comparison to transformer networks.

Following the suggestion of the reviewers, we have now added extensive new analyses and lesion studies to characterize the utility of the different components of our model (Extended Data Fig. 3). We found that, although the impact of the two behavioral modules (perspective and modulation) had a significant effect in performance, the effect was small (Extended Data Fig. 3a–j). Specifically, removing the perspective module resulted in 2.3% reduction in predictive accuracy, while removing the modulation module led to a 2.8% reduction. Similarly, we also found an even smaller but still significant advantage of using 3D convolutions in the feedforward component of the core module, which resulted in significantly better model performance than with 2D convolutions, although the relative performance difference was small at 0.88% (Extended Data Fig. 3k–o). Finally, we showed that models trained with mean squared error loss are 9.6% worse than models trained with Poisson negative likelihood loss (Extended Data Fig. 3p–t).

To explore whether newer transformer architectures that could yield better performance, we developed a new recurrent network architecture that utilized the convolutional tokenization and attention mechanism from the Convolutional Vision Transformer (CvT), a transformer network that achieves state-of-the-art accuracy in image classification [8]. The CvT operation was combined with the long short-term memory (Lstm) network to produce the CvT-Lstm architecture. This was compared to the original Conv-Lstm architecture, which combined the convolution operation with the Lstm network. When trained on the full amount of experimental data, we did not find a significant difference between the new CvT-Lstm and original Conv-Lstm architectures. However, when trained on restricted amounts of experimental data, we found that Conv-Lstm outperformed CvT-Lstm (Extended Data Fig. 5). In summary, while individual components contributed modest but significant improvements in predictive performance, with training loss having the largest effect, the main driver of increased performance is largely attributable to the much larger dataset used for training - a property exhibited by ANNs in general and known as scaling laws, which have been the main driver of impressive improvements in large language models and other AI systems.

■ Detailed responses to the reviewers comments

Referee 1 (Remarks to the Author):

The authors show that an artificial neural network (ANN) can be trained to reproduce recorded calcium imaging data for over 70k neurons in the visual cortex of the mouse. Furthermore, the ANN can predict very well the neural responses to visual stimuli that were not included in the training set. Substantial improvements over similar results from the same and other labs were achieved by including data on the current perspective of the mouse, inferred from eye-tracking data, and its behavioral state, inferred from locomotion and pupil dilation.

These results are certainly very nice and useful. But I do not find them so significant that they merit a publication in Nature. The insight which they provide into the structure or function of the visual cortex is limited. For example, it remains unknown how the features which the ANN extracts from the visual stimuli and the other recorded data are related to features that are extracted by the visual cortex. The authors make no effort to advance our insight on that. One underlying problem is that the recorded response of each neuron is predicted by a readout, a trained weighted sum of large numbers of features that the ANN extracts. These readout weights are adapted individually for each neuron. One cannot expect that these weighted sums have any relation to the synaptic input which the neuron receives in the brain, especially if one uses readouts from dozens or more units of the ANN. Hence in a sense, one just replaces one black box, the visual cortex, by another black box, the trained ANN. But since the trained ANN provides a new representation of the recorded data, it would be of interest to see whether they actually provide a compression of the recorded data. But this is not pursued in this study.

We thank the reviewer for their insightful comments, which raise important questions about the utility of data-driven deep neural network models in advancing neuroscience. We acknowledge that accurately modeling the brain functionally does not alone suffice for deeper understanding; further analysis of the model is crucial. In response to the reviewer's comments, and as we detail above, we have conducted additional experiments and analyses of the model that provide new insights into the visual cortex. In summary, as the reviewer suggested, we now demonstrate that the ANN model offers a useful, compressed representation of the complex nonlinear input-output functions of neuronal responses to natural movies. This nonlinear response function for each neuron is captured by what we term a "functional barcode," consisting of the readout weights from each feature map in the ANN's final convolutional layer. We found that these functional barcodes can predict the morphological cell types of excitatory neurons across layers 2-5 in the functional connectomics MICrONS dataset. Furthermore, we demonstrate the utility of our model and the functional barcodes in four other studies [5, 4, 6, 7], making novel discoveries about the

structure and function of cortical circuits, as discussed above. For example, our model enabled [5] to reveal that neurons primarily select synaptic partners based on what they respond to (feature selectivity) rather than where they respond (spatial selectivity), providing unique insight into the principles governing cortical connectivity. Moreover, in [7] our model provided empirical evidence for a Hubel and Wiesel like hierarchy analogous to their simple-to-complex cell model. This finding not only validates a long-standing hypothesis in neuroscience about how invariant representations are built, but also demonstrates how our approach bridges data-driven modeling with specific hypotheses about neural computation.

I was missing quantitative data on the architecture of the ANN model; I did not even find how many parameters are trained in a readout for a specific neuron. One could also debate whether the term “foundation model” is adequate for this trained ANN. Most currently discussed foundation models such as GPT or BERT, are based on transformers rather than standard ANNs. Apparently it remains unknown whether these newer types of models can provide further improvements for predicting large-scale neural recordings from the brain.

We apologize for not describing our model in more details. We now have added all the relevant details in the results and methods sections, including the dimensionality of the readout weights (*Readout module* section of *Methods*), as well as the number of layers and specific formulations of the network (*Perspective module*, *Modulation module*, and *Core module* sections of *Methods*). We also trained transformer-based architectures as suggested by the reviewer. We find that they perform as well the original Conv-Lstm architecture with full data, but not as well with limited data (see **Lesion studies and comparison to transformer networks** above). This may be due to a better inductive bias of the Conv-Lstm compared to the transformer-based Cvt-Lstm for the modeling dynamic neural activity. Alternatively, it could be due insufficient optimization of the transformer architecture on this data domain, and additional development may yield improved performance. Nevertheless, there is increasing evidence that the specific architecture used for large-scale machine learning is not as important as once believed [9]. Rather, it is the sheer magnitude of high-quality data that underlies the impressive capabilities of foundation models. We will also release the full code and parameters to reproduce the ANN model upon publication.

Referee 2 (Remarks to the Author):

In this paper, Wang et al. present a deep neural network model trained to predict the responses of neurons in mouse visual cortex to natural movies. The model consists of four components: a perspective network for transforming images into retinal activations, a modulation network for conditioning the responses on the animal’s behavior, a core network for transforming retinal activations into a latent space via 3D convolutions and recurrence, and a readout network for transforming the latent representation into neural activations. The authors show that they can train on data from multiple animals using the same core network, which then comprises something the authors present as being akin to a “foundation model” for predicting neural responses, where the term “foundation model” as used in the field of AI indicates a pre-trained backbone that can be fine-tuned on new downstream tasks. Here, it is shown that if they pre-train the foundation core with multiple animals then train on new animals’ data they can achieve better correlations between predicted activity and real activity than they achieve by training the model on each individual animal, particularly for low amounts of training data. They also show that the model can generalize to new stimuli and new types of stimuli, e.g. artificial stimuli rather than natural movies. The authors go on to show that thanks to the improved response predictions the model can be used to estimate basic properties of each neuron, including orientation selectivity, direction selectivity, spatial selectivity, and retinotopic position, which they verify using data from the MICrONS dataset.

In general, this paper is well-written and the authors have achieved an impressive technical feat. This model is achieving something that past models could not, namely, accurate prediction of responses to movies and generalization to new stimulus types. That is commendable and a clear contribution. However, evidence that this is really a novel approach or that it constitutes a “foun-

dation model” is actually quite limited, and the demonstrations provided of the model’s utility are a little bit underwhelming given how impressive the model is otherwise. As well, given that this paper ultimately presents a methodological advance, i.e. no novel information is learned about mouse visual cortex, it is important to provide more technical information on the development of the model than is provided currently. Below, I expand on these points, which I believe would need to be addressed to make this paper appropriate for publication in Nature. I then provide a few minor comments/questions.

Specifics

What is the novel approach here, and is this really a “foundation model”?

As noted in the paper, other papers (e.g. Lurz et al. (2020)) previously showed a transfer learning effect from training on one group of animals and then fine-tuning on another, so the major jump here seems to be using 3D convolutions and recurrence to make predictions on movies rather than still images. That is not a trivial engineering achievement, but is it really that novel an approach? In line with this concern, the implication is that this work is novel because it presents a “foundation model”. But, one of the most interesting aspects of traditional foundation models in AI, like large language models, is that they not only generalize to new data but also to new *tasks*. In contrast, here, the model is being transferred to the same task (predict neural responses from movies), and it is simply generalizing to new data in that same task. Could the authors provide an example of generalization to a novel task? Without this, it is arguably not a “foundation model”.

For example, could the model be fine-tuned to predict the sub-type of cells (e.g. pyramidal neuron, Martinotti cell, etc.) or which cells are synaptically connected to one another? These things could be done by taking the foundation core and the existing read-out layer for activity then adding a new secondary classification read-out layer that receives the predicted activity (and also potentially the latent representations) and outputs cell type or a set of connection probabilities. This could then be fine-tuned on the structural information from the MICrONS dataset, which bizarrely enough, is used only for its recordings and positional information and not its rich structural information, despite the hint at the beginning of the paper of this possibility. In fact, as far as I can tell, nothing is done to actually use the EM component of the MICrONS data here, which is an odd choice given that it is highlighted in the abstract and Figure 5. Hence, the downstream task of predicting cell types or synaptic connections seems like an obvious choice as a novel task to fine-tune on. But, if the authors can think of another task that is fine. The important thing is simply to show generalization to a novel domain other than predicting neural responses. Without that this isn’t a “foundation model”, but rather, just a pre-trained model being used for transfer learning, as has been done before.

We thank the reviewer for this comment and agree that the utility of our model, and its qualification as a foundation model, would be significantly strengthened if we could demonstrate its performance on new tasks beyond neural prediction. Following these suggestions, we have now conducted extensive new analyses and demonstrate in this paper, as well as in four companion papers, its capabilities in predicting morphological cell types, dendritic features, and synaptic connectivity, which we discuss in detail in our *General response* above.

What could we actually use a model like this for other than reducing the recording time required in experiments? What could we learn?

Though the engineering feat they have engaged in is impressive, there is relatively little done with it in the way of learning something new. What is shown in this paper is that the model can be used to predict simple tuning curve properties that can easily be measured with more recording time, so the model could theoretically save on recording time in experiments. But, is that really a major advance worthy of publication in Nature? Arguably, the authors should at least provide some speculation about what else we could actually learn from such a model, and if possible, even demonstrate or confirm one piece of such speculation. Could we predict previously unknown functional cell types? Could we predict learning trajectories to verify theories of learning

rules? What can be done other than simply saving time in experiments? This is important for demonstrating that this work is something more than a simple extension of previous models for fitting neural responses to the temporal domain. Even some speculation here would help strengthen the paper.

We thank the reviewer for their suggestions. Following these criticisms, in addition to using our model to predict morphological cell types, we have also applied it in companion papers that analyze the MICrONS dataset to decipher the relationships between structure and function of cortical circuits, which are now summarized above.

In particular, the ability of our model to factorize the tuning function of these neurons into a 'feature what' and 'spatial where' component enables us to gain a much more granular understanding of the rules that govern the wiring of neurons. In the companion paper by Ding et al. [5], which used our model, they show that the feature component, but not the spatial component, predicted which neurons were connected at the fine synaptic scale. We agree that there are many exciting directions to analyze our dynamic model and apply AI interpretability tools. Although these tools are being developed at a rapid pace, they are just beginning to be applied to dynamic models. We are very interested in developing these methods and are actively pursuing this direction. However, we believe this would require substantial new effort and a series of closed-loop experiments to verify these discoveries in the brain, going beyond the scope of the current manuscript. In summary given these new additions, our model has already demonstrated significant utility beyond simply saving experimental time.

How were the design choices for the network arrived, e.g. how were hyperparameter and architecture choices made?

Given that this is largely a methodological paper, one critical piece missing is a more thorough description and analysis of the design choices made in the model and their impact. For example, some "lesion" studies should be included on the different components of the model. How important is the perspective or modulation network for performance? What if the loss function at the output is changed? What if 2D convolutions are used instead of 3D convolutions? Similarly, how were hyperparameter choices made (learning rate schedule, batch size, momentum, etc.)? Why was the particular optimizer that was used selected? And so on...

In the absence of these kind of details it will be more difficult for other scientists to learn from the experience here and expand on this model to build new models for similar purposes. Hence, these sorts of matters are very important for a paper like this.

We agree that these lesion studies would add value to our paper. We now report these in Extended Data Fig. 3, where we lesion specific components of the model and observe the effect of the lesion on model performance. We evaluated the effect the perspective module, the modulation module, convolution type (2D vs 3D), and loss function (Poisson negative log likelihood vs mean square error). Please also see the "Lesion studies and comparison to transformer networks" section above.

We have added a "Model hyperparameters" section to the Methods, where we explain how we chose the architecture and training hyperparameters (e.g., learning rate, batch size, momentum, etc.) were chosen. Briefly, to identify optimal hyperparameters, we performed a grid search using a preliminary dataset. For the final results and figures, we trained new models on a separate dataset, using the hyperparameters identified in the grid search. To prevent hyperparameter overfitting, there was no overlap in animal subjects between the dataset used for the grid search and the dataset used for the final results.

Minor comments

Why have the different "layers" in the recurrent nets in the core? If they're all connected to each other bidirectionally, is this not just equivalent to having one recurrent network with three times the number of channels? Is there something I'm missing?

We have now changed the terminology of "layers" to "cells" to avoid the connotation of directionality associated with "layers". Our implementation of the recurrent network makes it slightly

different from having multiple sets of channels simply stacked together to form one large vanilla recurrent network. Instead, there is a slight difference in how channels from different cells are processed with respect to each other. Channels within each cell are connected by 3×3 convolutions, whereas channels across cells are connected by 1×1 convolutions. For more details, please see the "Core module" section of the Methods.

Does Figure 5b do anything? The EM data is not used here at all as far as I can tell. It is weird to include an image of it and then do nothing with it.

We have now done the analysis suggested by the reviewer predicting anatomically defined cell types in the MICrONS data set from [3] linking our model to the EM data and have changed Figure 5 accordingly.

For the OSI and DSI estimates, how does the reliability of the model compare to the reliability one would get from multiple experiments in the same animal?

We have now added Extended Data Fig. 6 that shows the inter-experimental reliability of the *in vivo* and *in silico* estimates of DSI, OSI, and preferred direction and orientation. We computed these metrics for neurons that were recorded more than once in the MICrONS volume (due to overlaps in brain regions between recording sessions). For each recording session, we computed the tuning metrics (DSI, OSI, preferred direction and orientation) and display them as scatter-plots for both *in vivo* and *in silico* estimates (Extended Data Fig. 6 a-b, d-e, g-h, j-k). We also show that, across the recorded neurons, the difference in tuning metrics across experiments is similar when comparing *in silico* and *in vivo* approaches (Extended Data Fig. 6 c, f, i, l).

It took me a while to understand what was being shown in Extended Data Figure 4(now 7)a-c. I think there might be a way to make it easier to understand, e.g., by simplifying the axes labels or providing a bit more explanation of what the data shows in the caption.

We edited the caption of the figure to provide a more straightforward explanation of what the data shows in panels a–c.

References

- [1] Konstantin-Klemens Lurz et al. "Generalization in data-driven models of primary visual cortex". In: *International Conference on Learning Representations*. 2021.
- [2] Fabian H. Sinz et al. "Engineering a Less Artificial Intelligence". In: *Neuron* 103.6 (Sept. 2019), pp. 967–979. DOI: 10.1016/j.neuron.2019.08.034. URL: <https://doi.org/10.1016/j.neuron.2019.08.034>.
- [3] Casey M Schneider-Mizell et al. "Cell-type-specific inhibitory circuitry from a connectomic census of mouse visual cortex". In: *bioRxiv* (2023).
- [4] Marissa A. Weis et al. *An unsupervised map of excitatory neurons' dendritic morphology in the mouse visual cortex*. Dec. 22, 2022. DOI: 10.1101/2022.12.22.521541.
- [5] Zhuokun Ding et al. *Functional connectomics reveals general wiring rule in mouse visual cortex*. preprint. Neuroscience, Mar. 14, 2023. DOI: 10.1101/2023.03.13.531369. URL: <http://biorxiv.org/lookup/doi/10.1101/2023.03.13.531369>.
- [6] Jiakun Fu et al. *Pattern completion and disruption characterize contextual modulation in mouse visual cortex*. preprint. Neuroscience, Mar. 14, 2023. DOI: 10.1101/2023.03.13.532473. URL: <http://biorxiv.org/lookup/doi/10.1101/2023.03.13.532473>.
- [7] Zhiwei Ding et al. *Bipartite invariance in mouse primary visual cortex*. preprint. Neuroscience, Mar. 16, 2023. URL: <http://biorxiv.org/lookup/doi/10.1101/2023.03.15.532836>.
- [8] Haiping Wu et al. *CvT: Introducing Convolutions to Vision Transformers*. Mar. 29, 2021. arXiv: 2103.15808[cs]. URL: <http://arxiv.org/abs/2103.15808>.
- [9] Samuel L. Smith et al. *ConvNets Match Vision Transformers at Scale*. Oct. 25, 2023. arXiv: 2310.16764[cs]. URL: <http://arxiv.org/abs/2310.16764>.